



# GCAP 2.0: A global 3-D chemical-transport model framework for past, present, and future climate scenarios

Lee T. Murray[1,2], Eric M. Leibensperger[3], Clara Orbe[4], Loretta J. Mickley[5], and Melissa Sulprizio[5]

[1]Dept. of Earth and Environmental Sciences, University of Rochester, Rochester, NY USA
[2]Dept. of Physics and Astronomy, University of Rochester, Rochester, NY USA
[3]Dept. of Physics and Astronomy, Ithaca College, Ithaca, NY USA
[4]NASA Goddard Institute for Space Studies, New York, NY USA
[5]School of Engineering and Applied Sciences, Harvard University, Cambridge, MA USA

**Correspondence:** Lee T. Murray (lee.murray@rochester.edu)

**Abstract.** This manuscript describes version 2.0 of the Global Change and Air Pollution (GCAP 2.0) model framework, a one-way offline coupling between version E2.1 of the NASA Goddard Institute for Space Studies (GISS) general circulation model (GCM) and the GEOS-Chem global 3-D chemical-transport model (CTM). Meteorology for driving GEOS-Chem has been archived from the E2.1 contributions to Phase 6 of the Coupled Model Intercomparison Project (CMIP6) for the preindustrial and recent past. In addition, meteorology is available for the near future and end-of-the century for seven future scenarios ranging from extreme mitigation to extreme warming. Emissions and boundary conditions have been prepared for input to GEOS-Chem that are consistent with the CMIP6 experimental design. The model meteorology, emissions, transport and chemistry are evaluated in the recent past and found to be largely consistent with GEOS-Chem driven by the Modern-Era Retrospective analysis for Research and Applications Version 2 (MERRA-2) product and with observational constraints.

## 1 Introduction

How atmospheric composition and chemistry will change in the future and has changed in the past is of tremendous societal importance. Surface air pollution is the leading cause of preventable death worldwide (GBD 2019 Risk Factor Collaborators, 2020) and threatens global food security and ecosystem health (e.g., Tai et al., 2021). Short-lived climate forcers like methane, tropospheric ozone, and aerosol particles influence global and regional climate (e.g., Fiore et al., 2015). The emission of ozone-depleting substances and climate change threatens the overlying stratospheric ozone layer (WMO, 2018). And solar radiation modification through the purposeful injection of chemical species into the atmosphere is seriously being considered to combat anthropogenic climate change. Yet, it remains highly uncertain in its efficacy or risks (e.g., Eastham et al., 2021).

To study and address these issues for the recent past, scientists, regulators, and policymakers frequently use 3-D chemical-transport models (CTMs). CTMs use archived meteorology to drive the spatial and temporal evolution of trace gases and particles in the atmosphere. By not needing to resolve all the equations of motion, as in a general circulation model (GCM), CTMs can expend additional computational power to resolve more complex chemistry or perform additional simulations. Furthermore, the chain of cause and effect between meteorology and composition is much easier to establish in a CTM than a





fully coupled chemistry-climate model (CCM) or Earth-system model (ESM) running online chemistry. And because meteorological reanalyses usually drive CTMs, they may easily be matched in space and time to observations, unlike CCMs or

ESMs that generate their own winds. However, the reliance of CTMs on existing driving meteorology means that CTMs have traditionally been largely excluded from international assessments aimed at forecasting future or past changes, such as those of the ongoing Phase 6 of the Coupled Model Intercomparison Project (CMIP6, Eyring et al., 2016) that is set to inform the upcoming Intergovernmental Panel on Climate Change (IPCC) Sixth Assessment Report.

Here we introduce, describe and evaluate version 2.0 of the Global Change and Air Pollution (hereafter, "GCAP 2.0")

chemical-transport model framework. GCAP 2.0 represents a major update and expansion of the original GCAP described by Wu et al. (2007) and Murray et al. (2014). Meteorology necessary for driving the grassroots-community GEOS-Chem 3-D chemical-transport model (http://www.geos-chem.org) has been archived for the preindustrial, recent past and several future scenarios of the CMIP6 experiment using version E2.1 of the NASA Goddard Institute for Space Studies (GISS) GCM (Kelley et al., 2020; Miller et al., 2021). In addition, the CMIP6 emissions and surface boundary conditions have been processed for

use within GEOS-Chem for consistency with the driving meteorology and to enable GEOS-Chem to perform and contribute to the CMIP6 experiments.

Section 2 summarizes the history of the GCAP framework. Section 3 describes the climate and chemistry models used and their interface. Section 4 summarizes and evaluates the meteorology products in the recent past versus reanalyses. Section 5 describes the emissions and boundary conditions and evaluates the climate-sensitive emissions. Section 6 evaluates the model

in the recent past by comparing it to observations. We conclude with a summary section.

## 2 History

The GISS GCM and GEOS-Chem CTM have a long history of collaborative development. The immediate predecessor to GEOS-Chem was a gas-phase CTM of tropospheric ozone-$NO_x$-CO-hydrocarbon chemistry (Wang et al., 1998a, b, c; Wang and Jacob, 1998) driven by present-day meteorology archived from version II' of the GISS GCM at 4° latitude by 5° longitude

horizontal resolution with seven vertical layers extending from the surface to 150 hPa (Hansen et al., 1983; Rind and Lerner, 1996). GEOS-Chem was born when Bey et al. (2001) updated this model to include the tropospheric non-methane hydrocarbon oxidation mechanism of Horowitz et al. (1998) and allowed it to be driven instead by the Goddard Earth Observing System (GEOS) assimilated meteorological reanalyses produced by the NASA Global Modeling and Assimilation Office (GMAO) (Schubert et al., 1993). This early version of GEOS-Chem incorporated the GEOS dynamical core (Lin and Rood, 1996).

Shortly afterward, a bulk sulfur-nitrate-ammonium aerosol mechanism was included by Park et al. (2004).

Since these origins, GEOS-Chem has developed a large user and active developer base of hundreds of individuals at more than 150 institutions in over 30 countries (http://www.geos-chem.org). The model is extensively versioned, documented, and benchmarked. Other subsequent major developments include the development of one- and two-way coupled nested regional simulations (Wang et al., 2004; Yan et al., 2016; Bindle et al., 2020), an adjoint for inverse model applications (Henze et al.,

2007; Kopacz et al., 2009), a unified chemical mechanism from the surface to the mesopause (Eastham et al., 2014), a flexible





emissions pre-processor (Keller et al., 2014), and a massively parallel distributed computing framework enabling global simulations down to resolutions of 0.25° latitude by 0.3125° longitude with 72 vertical layers extending to 0.01 hPa (Eastham et al., 2018).

The Global Change and Air Pollution (GCAP) framework developed by Wu et al. (2007) re-enabled version 7-02-04 of GEOS-Chem to be driven by GISS meteorology in order to explore how changes in future climate and precursor emissions may influence surface air quality (e.g., Wu et al., 2008a, b; Pye et al., 2009; Pye and Seinfeld, 2010; Selin et al., 2009; Hui and Hong, 2013; Zhu et al., 2017). GCAP utilized meteorology archived from version III of the GCM (Rind et al., 2007) at 4° latitude by 5° longitude resolution with 23 vertical layers extending to 0.002 hPa for the present-day and the Intergovernmental Panel on Climate Change (IPCC) Special Report on Emissions Scenarios (SRES) "A1B" scenario for 2050 C.E. (Nakicenovic and Swart, 2000).

In the subsequent ICe Age Chemistry And Proxies (ICECAP) project, Murray et al. (2014) updated the GCAP implementation to enable version 9-01-03 of GEOS-Chem to be driven by paleo meteorology archived from version E of the GISS GCM (Schmidt et al., 2006) and consistent land cover simulated using terrestrial vegetation models (Kaplan et al., 2006; Pfeiffer et al., 2013) to explore chemistry-climate changes at and since the Last Glacial Maximum (LGM; ∼21 kyr before present; Murray et al., 2014; Achakulwisut et al., 2015; Geng et al., 2015, 2017).

However, the original GISS-driven variants of GEOS-Chem suffered from several issues. Most notably, the stratosphere-to-troposphere mass flux was always too large, complicating the tropospheric ozone budget and the interpretation of polar ice-core records once GEOS-Chem developed online interactive stratospheric chemistry. The use of the more accurate, but computationally expensive, GISS dynamical core within GEOS-Chem to improve transport yielded severe performance issues in the CTM. At the time, both GEOS-Chem and the GISS GCM used their own in-house binary formats for file input and output that required translation (versus the standard NetCDF file format used today by both models). Lastly, the different horizontal and vertical resolutions required extensive offline processing of input fields. GCAP was eventually deprecated and removed from the GEOS-Chem codebase in version 11-02d.

However, subsequent developments to both models increased flexibility and capabilities, motivating the development of GCAP 2.0, as described in the subsequent section.

## 3  Model Description

GCAP 2.0 is the second generation of a one-way offline coupling between the NASA GISS GCM and the GEOS-Chem CTM. Meteorology archived from version E2.1 of the GCM for any period of Earth history or its future may be used to drive the GEOS-Chem CTM. The following subsections describe the salient components and edits to the GCM and CTM relevant for GCAP 2.0 simulations.

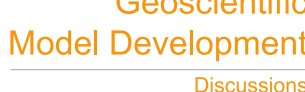
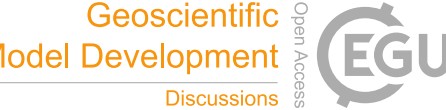

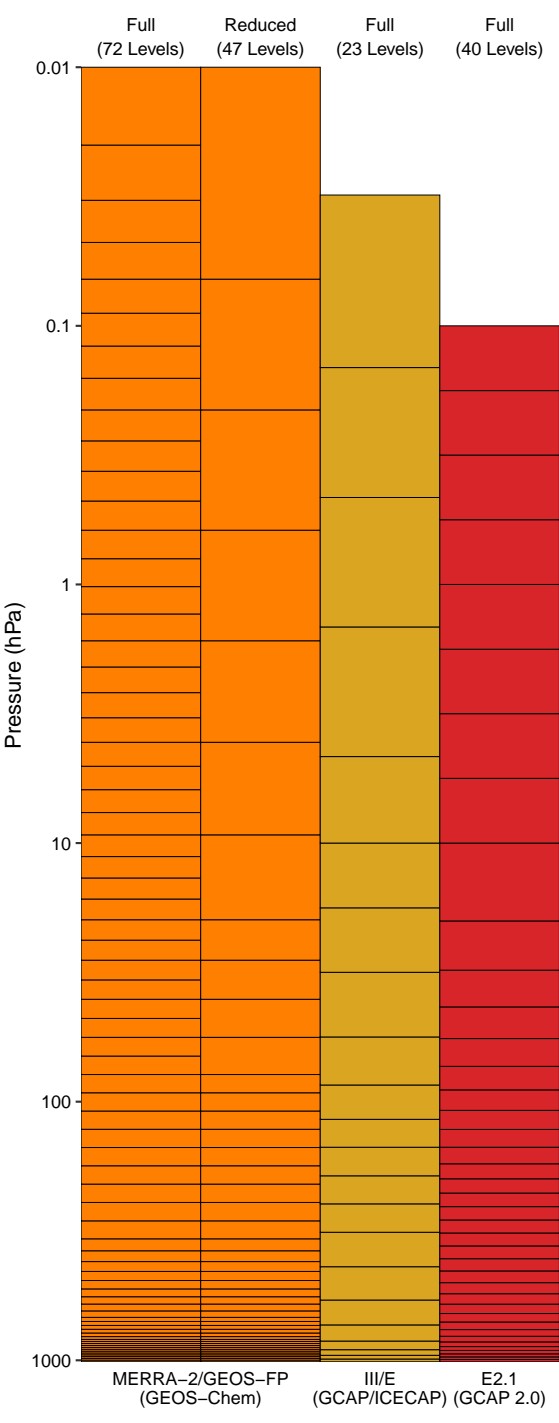

**Figure 1.** Comparison of the vertical resolutions of GEOS-Chem driven by the MERRA-2 or GEOS-FP reanalyses (full and the reduced stratospheres; orange), the original GCAP driven by Model III or ICECAP driven by ModelE (yellow), and GCAP 2.0 driven by E2.1 (red).



### 3.0.1 NASA GISS ModelE2.1

The version of the GISS GCM frozen and applied to the initial CMIP6 experiments (ModelE2.1, hereafter "E2.1") is described in detail by Kelley et al. (2020) and Miller et al. (2021). In brief, the standard E2.1 configuration resolves the equations of mass, momentum, and energy in Earth's atmosphere at a horizontal resolution of 2° latitude by 2.5° longitude and with 40 vertical layers extending from the surface to 0.1 hPa (∼28 in the tropical troposphere; see Fig. 1). The model employs a quadratic-upstream scheme for advection that yields finer effective spatial resolutions by transporting higher-order moments of the sub-grid distributions (Prather, 1986). Gravity-wave momentum fluxes resulting from flow over topography and fronts are parameterized as stratospheric drag processes (Rind et al., 1988). Moist convection underwent substantial updates relative to E2.1's predecessor version, E2 (Kim et al., 2012; Del Genio et al., 2012, 2015). Radiation physics includes calculations for major shortwave and longwave absorbers (water vapor, carbon dioxide, ozone, methane, nitrous oxide, chlorofluorocarbons) and aerosol particles (Hansen et al., 1983), any of which may be either prescribed or calculated online as a function of emissions, chemistry, and physical losses. The influence of aerosol particles on cloud microphysics and albedo may be explicitly represented or parameterized (Bauer et al., 2020). The model may be coupled to a fully interactive ocean model or be applied in atmosphere-only mode through prescribed sea-surface temperatures.

GISS contributed several configurations of E2.1 to CMIP6. Here, we use the atmosphere-only configuration with composition prescribed from earlier runs using online and interactive chemistry for computational expediency. At the time of publication, GISS also contributed up to 11 ensemble members per historical and future emission scenario initialized from different moments of the preindustrial control simulation. We focus on the atmosphere-only ensemble member that contributed to the largest number of Tier 1 and Tier 2 scenarios of the CMIP and Scenario Model Intercomparison Project (ScenarioMIP) experiments, corresponding to variant label "r1i1p1f2" in the CMIP6 data repository (https://esgf-node.llnl.gov/projects/cmip6/).

On top of the E2.1 codebase used for the CMIP6 simulations, we implemented new sub-daily diagnostics that archive the same fields used to drive GEOS-Chem as generated by the Modern-Era Retrospective analysis for Research and Applications Version 2 (MERRA-2) meteorological reanalysis (Gelaro et al., 2017). We then re-performed the "r1i1p1f2" variant of the E2.1 contributions to the CMIP and ScenarioMIP experiments using initial and intermediate restart files archived during the original simulations and archiving the sub-daily diagnostics necessary for driving GEOS-Chem. We discuss and evaluate the meteorology in Sect. 4. Three-dimensional fields were archived at 3 h temporal resolution and two-dimensional fields were archived at hourly temporal resolution, consistent with the MERRA-2 product. In addition, we archived hourly lightning flash densities and convective cloud depths. The only difference in the repeat simulation configurations with respect to their original runs was a need to call the radiation code every dynamic time step instead of every five to obtain the hourly radiation fields necessary for driving GEOS-Chem; the consequences of this are discussed in Sect. 4.

Table 1 summarizes the 178 years of GCAP 2.0 meteorology archived and publicly available at publication time. For comparison, MERRA-2 presently has 41 years of complete meteorology available. The data are publicly served from the new GCAP data server hosted by the University of Rochester Atmospheric Chemistry and Climate Group at http://atmos.earth.rochester.edu/input/gc/ExtData/. Users can point to this repository analogously to the existing GEOS-Chem data servers hosted by Harvard (https://ftp.as.harvard.edu/gcgrid/data/ExtData/)



**Table 1.** GCAP 2.0 meteorology available at time of publication from the GCAP 2.0 data repository.

| Scenario | Variant Label | 1851-1860 | 2001-2014 | 2040-2049 | 2090-2099 |
|---|---|---|---|---|---|
| Historical | r1i1p1f2 | X | X | | |
| Historical (Nudged to MERRA-2) | | | X | | |
| SSP1-1.9 (Extreme Mitigation) | r1i1p1f2 | | | X | X |
| SSP1-2.6 | r1i1p1f2 | | | X | X |
| SSP4-3.4 | r1i1p1f2 | | | X | X |
| SSP2-4.5 | r1i1p1f2 | | | X | X |
| SSP4-6.0 | r1i1p1f2 | | | X | X |
| SSP3-7.0 | r1i1p1f2 | | | X | X |
| SSP5-8.5 (Extreme Warming) | r1i1p1f2 | | | X | X |

All meteorology fields available at 2° latitude by 2.5° longitude with 40 vertical layers from the surface to 0.1 hPa. Two-dimensional fields are archived at hourly temporal resolution. Three-dimensional fields are archived at 3 h temporal resolution.

or Compute Canada (http://geoschemdata.computecanada.ca/ExtData/). Historical meteorology has been archived for the prein-
dustrial (1851-1860 C.E.) and recent past (2001-2014 C.E.). In addition, we archive near-future (2040-2049 C.E.) and end-of-
the-century (2090-2099 C.E.) meteorology for seven future scenarios ranging from extreme mitigation to extreme warming
(see Sect. 5 for a description of the emission scenarios). In addition, to facilitate comparison of GCAP 2.0 meteorology and
composition with observations and traditional GEOS-Chem, we have also performed a recent past simulation in which the
E2.1 horizontal winds of the r1i1p1f2 variant were "nudged" to match those of the MERRA-2 reanalysis for 2001-2014 C.E.
(Menon et al., 2008). Note that we only nudge the winds and not temperature, humidity or surface pressure as may be done in
other models. We urge users to be aware of the challenges involved when interpreting the impact of nudged meteorology on
atmospheric composition, especially in the stratosphere (e.g., see Orbe et al., 2020a).

### 3.0.2 GEOS-Chem

GEOS-Chem (http://www.geos-chem.org) is a global or regional 3-D chemical transport model traditionally driven by as-
similated meteorology products produced by the NASA Global Modeling and Assimilation Office (GMAO) Goddard Earth
Observing System Data Assimilation System (GEOS-DAS). The MERRA-2 science product is generated at 0.5° latitude by
0.625° longitude and 72 vertical layers extending from the surface to 0.01 hPa (~38 layers in the tropical troposphere) and
available from 1980 C.E. to the present (Gelaro et al., 2017). There is also a near real-time product (GEOS-FP) available at
0.25° latitude by 0.3125° longitude horizontal resolution and available from 2012 C.E., although with periodic changes to the
underlying code. Both products are provided at hourly temporal resolution for two-dimensional fields and at 3 h resolution for





three-dimensional fields[1]. Most GEOS-Chem users make use of these fields that have been pre-processed to coarser horizontal resolutions (4° latitude by 5° longitude or 2° latitude by 2.5° longitude)[2] for computational expediency and to minimize storage requirements. Users may also select to run with reduced vertical resolution in the stratosphere (see Fig. 1).

Emissions are the subject of Sect. 5. The original description of the tropospheric chemical mechanism is by Bey et al. (2001), which was expanded to include a stratospheric mechanism by Eastham et al. (2014), the latter of which did not exist in the earlier versions of GCAP. The coupled sulfur–nitrate–ammonium aerosol simulation is described by Park et al. (2004) with aerosol thermodynamics computed via the ISORROPIA II model (Fountoukis and Nenes, 2007). Advection is handled by a flux-form and partially semi-Lagrangian transport scheme (Lin and Rood, 1996). Convective transport is parameterized as a

single plume acting under the mean upward convective, entrainment and detrainment mass fluxes for each level of a model column as archived from the GCM.

GEOS-Chem developed the capability to be driven by any horizontal resolution beginning with the "FlexGrid" update in version 12.4.0 (doi:10.5281/zenodo.3360635). The model can now define any horizontal resolution at initialization and automatically re-grid input meteorology upon file read from its archived resolution to the run-time resolution. Because our strategy

was to archive from E2.1 the same fields as in the MERRA-2 reanalysis, very few modifications were necessary to the GEOS-Chem source code to allow GEOS-Chem to use E2.1 output as a meteorological driver. The primary additional code required is the inclusion of the specification of the E2.1 vertical resolution. Otherwise, the only other GEOS-Chem changes were removing some hard-coded limitations, e.g., those that prevented the model from running on dates before Jan 1, 1900. These updates are scheduled to enter the standard GEOS-Chem code in version 13.1 (doi:TBD; will be known by publication). Version 13.0.0 of

GEOS-Chem (doi:10.5281/zenodo.1343546) introduced the ability of the source code to generate run directories, and we have also submitted code updates to version 13.1 to do so for GCAP 2.0. We have re-gridded all restart files to the new 40-layer vertical resolution.

Because of the relatively few required changes, GCAP 2.0 meteorology is easily compatible with any existing GEOS-Chem capability. There are three primary methods by which GEOS-Chem may be used. The first and most common method due

to its ease of installation and application is GEOS-Chem Classic or "GCClassic." Therefore, we have guaranteed that all GCClassic configurations work with GCAP 2.0 by including run directories and regridding all input files that have a vertical dimension. In addition to full chemistry simulations, these include the speciality simulations (e.g., offline aerosol, methane, carbon dioxide, tagged CO, tagged methane, tagged ozone). Only the existing GCClassic mercury simulation will require some modifications; in the interest of storage, we did not archive the ten extra sea-ice fields only used by that simulation since they

may be determined online from the fraction of sea ice field that was archived. FlexGrid also enables one to perform a global simulation at a relatively coarse resolution to archive boundary conditions for driving nested regional simulations at higher

---

[1]Like many free-running climate models, E2.1 uses a 365-day calendar, whereas GEOS-Chem includes leap days; the default behavior of GCAP 2.0 is to repeat Feb 28 meteorology on Feb 29. Users alternatively may stop the model at the end of Feb 28 and apply the restart file to Mar 1 for leap years to avoid meteorological discontinuities.

[2]Note that the native horizontal grid of E2.1 is offset from that traditionally used by GEOS-Chem at comparable resolutions. The former has the International Date Line as a cell edge whereas the later has it as a cell midpoint. In addition, E2.1 does not make use of half-polar cells as does GEOS-DAS or GEOS-Chem, making the total number of latitude bands one fewer in E2.1 as opposed to GEOS-Chem.

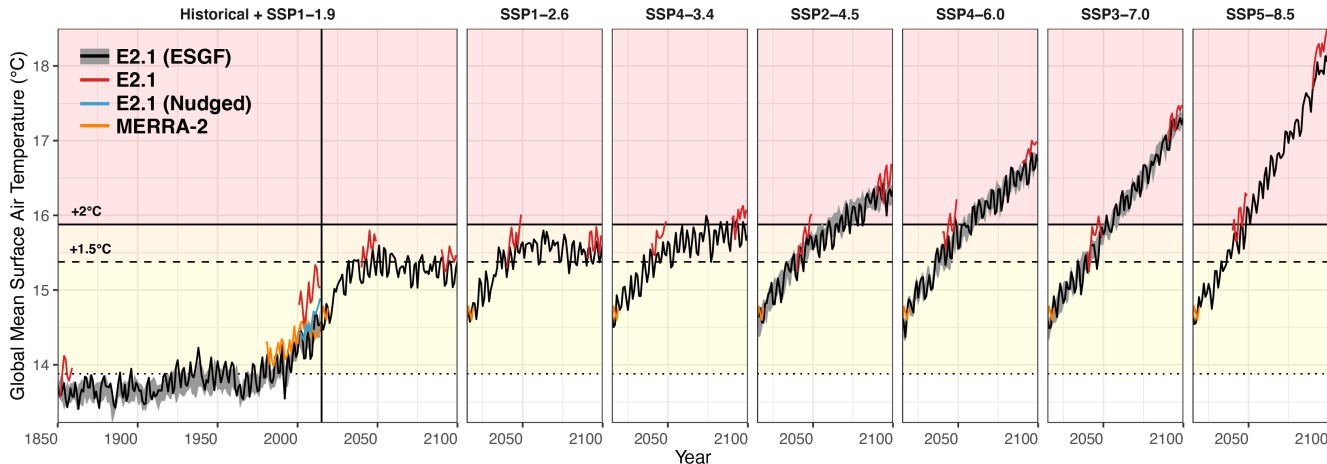

**Figure 2.** Temporal evolution of global mean surface temperature in °C. Each panel from left to right shows the seven future scenarios arranged by increasing radiative forcing. The black line shows variable *tas* from the E2.1 ensemble member (r1i1p1f2) that our simulations are based upon obtained from the ESGF repository. The gray shading represents the annual mean and 2-$\sigma$ spread of all E2.1 ensemble members archived on ESGF. The red line shows the value from our re-runs of r1i1p1f2 to generate the GCAP 2.0 meteorology. The blue line shows the same, but with winds nudged to the MERRA-2 reanalysis. The orange line shows the global mean surface temperature from the MERRA-2 reanalysis. The horizontal lines show the global preindustrial climatological mean in our simulations (dotted black) and 1.5°C (dashed black) and 2.0°C (solid black) increases on top of the preindustrial values.

spatial resolution. Although the underlying meteorology would still be the 2° latitude by 2.5° longitude of the GCAP 2.0 meteorology, one can benefit from the finer spatial resolution of the emission inventories.

The second method of running GEOS-Chem is the Message Passing Interface (MPI) parallelized variant utilizing a cubed-sphere dynamical core known as GEOS-Chem High-Performance (GCHP; Eastham et al., 2018). GCAP 2.0 meteorology is fully compatible with GCHP, although we refer the reader to Sect. 5.2 about the necessary pre-processing of emissions for GCAP 2.0 runs using GCHP.

Lastly, there exists an adjoint of GCClassic used for inverse modeling and sensitivity applications (Henze et al., 2007). Since the adjoint presently works with MERRA-2 meteorology, the GCAP 2.0 meteorology is also compatible with the adjoint code once the vertical resolution is added, allowing for inverse modeling applications in past and future climates.

## 4 Meteorology

This section evaluates the GCAP 2.0 meteorology by comparing it to its original CMIP6 simulation, the CMIP6 E2.1 ensemble, and the MERRA-2 reanalysis. Model output contributed to the CMIP6 experiment is archived by an international distributed data repository powered by the Earth System Grid Federation (ESGF) and available online at https://esgf-node.llnl.gov/projects/cmip6.

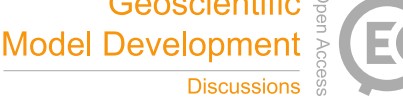

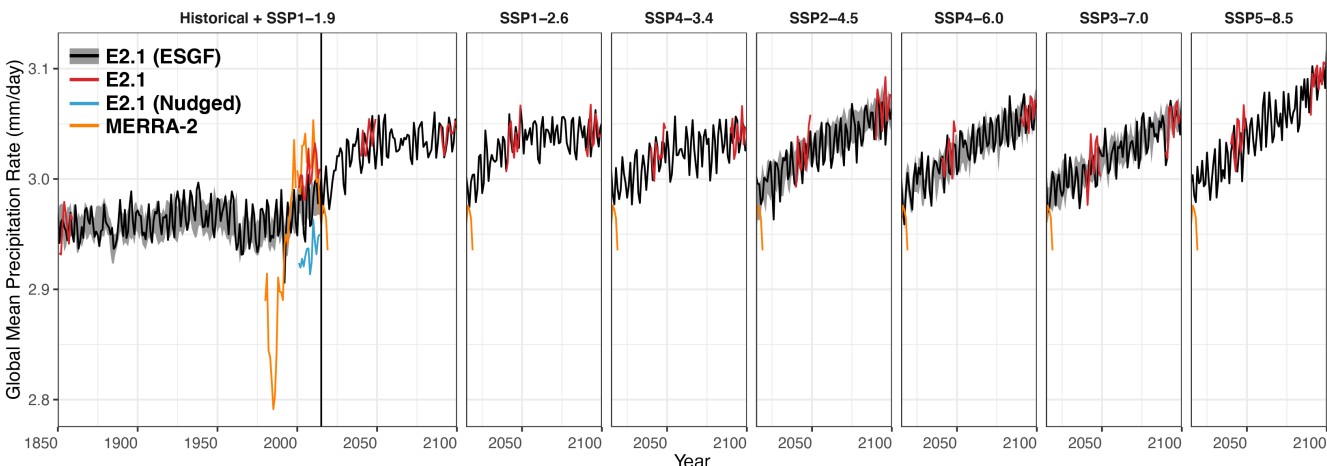

**Figure 3.** The same as Fig. 2, but showing the temporal evolution of the global mean precipitation rate in mm d$^{-1}$ (variable *pr* on ESGF).

Figure 2 shows the temporal evolution of annual mean surface air temperature. The black line shows the E2.1 r1i1pif2 variant and the gray shading shows the mean and 2-$\sigma$ spread of the submitted E2.1 ensemble from ESGF (ranging from 11 members in the historical to 1 member in some future scenarios). Our repeat simulations of the r1i1p1f2 variant from its archived restart files are shown in red and the same variant nudged to MERRA-2 is shown in blue. The MERRA-2 historical record is shown in orange.

First, we note that the repeat simulations have slightly warmer surface air temperatures than the original simulations. A small portion of this difference results from numerical noise associated with the original and repeat simulations' different computer architectures (NASA Center for Climate Simulations versus the University of Rochester Center for Integrated Research Computing). However, tests revealed that the increased frequency in calls to the radiation code necessary to archive hourly radiation fluxes for driving GEOS-Chem explains most of the difference in surface temperature. Although almost no locations show

significant changes with respect to local interannual variability (see Fig. S1 in the supplementary materials), this does lead to a weakly but statistically different annual mean temperature with respect to the original simulation (+0.6°C for 2001-2014 C.E., $p$-value = 0.002). However, the change is largely a linear offset, with temporal correlation remaining high ($R = 0.87$ for 2001-2014 C.E.), providing confidence in the ability of GCAP 2.0 to produce changes in composition associated with changes in climate accurately. Nudging the winds to MERRA-2 reduces this offset by influencing the rate of mixing of air between the

high- and mid-latitudes, and the warmer Arctic surface temperatures in our repeat free-running E2.1 simulations are in greater agreement with MERRA-2.

    Second, we note that for researchers interested in studying a "Paris Agreement"-like future world, in which future warming is limited to 2°C over preindustrial levels, one may use the SSP1-1.9, SSP1-2.6, or SSP4.3.4 scenarios. However, if one wishes to study a future world with the more aggressive goal of limiting warming to 1.5°C over preindustrial levels, then the only

scenario that may be used is SSP1-1.9.




Figure 3 shows the temporal evolution of the global annual mean precipitation rate in the simulations. Global precipitation rates are forecast to increase in the coming century due to the temperature-driven increase in surface evaporation and saturation vapor pressures. Unlike surface air temperature, the repeat simulations closely follow the original values and are statistically identical except in the recent past historical simulation, where they are globally higher by 0.9 %. The MERRA-2

reanalysis shows substantially more interannual variability in its global precipitation rates. Nudging E2.1 decreases the global precipitation rate.

Figures S2 to S51 of the supplemental materials include detailed comparisons of the seasonal climatologies for all fields in the three meteorology products that may be used to drive GEOS-Chem for 2005-2014 C.E. (MERRA-2, E2.1 nudged to MERRA-2, and the free-running E2.1). In general, most fields show excellent agreement with high pattern (i.e., spatial)

correlation and small mean difference. However, a few fields differ between E2.1 and MERRA-2 that are of interest for chemical-transport modeling, which we now summarize. The primary difference between MERRA-2 and E2.1 is the relative importance of stratiform versus convective precipitation. Whereas both models agree on the total precipitation flux to the surface (Fig. S16), MERRA-2 has a higher rate of stratiform condensation (Fig. S33) balanced by a higher rate of stratiform re-evaporation (Fig. S45). In contrast, E2.1 has a higher rate of convective condensation (Fig. S34) balanced by a higher rate

of convective re-evaporation (Fig. S44). Furthermore, E2.1 has consistently smaller surface roughness heights over the ocean and vegetated regions than MERRA-2; in contrast, MERRA-2 does not appear to include an orographic component in its surface roughness calculation and therefore has lower surface roughness heights over non-vegetated land surfaces (Fig. S29). Consequently, the E2.1 simulations have lower planetary boundary layer heights over oceans and heavily vegetated regions (globally ~200 m lower; Fig. S12) relative to MERRA-2. The E2.1 simulations also have a lower tropopause pressure by

approximately 40 hPa (Fig. S23); this has been corrected in version E2.2 of the GCM by moving to a higher vertical resolution (Orbe et al., 2020b). Lastly, E2.1 has a higher fraction of photosynthetically active radiation (PAR) present as diffuse (Fig. S10) rather than direct (Fig. S11) radiation, which will promote higher levels of biogenic emissions (see Sect. 5.2). Note also that MERRA-2 sets PAR fluxes to zero over water. Therefore, coastal and island cells will underestimate the radiation flux in MERRA-2-driven GEOS-Chem simulations, again with consequences for biogenic emissions.

Figure 4 compares the spatial distribution of key surface meteorological variables generated for GCAP 2.0 with their MERRA-2 counterparts. Surface air temperature shows near-perfect agreement in spatial distribution between the E2.1 products and MERRA-2. The E2.1 temperatures are slightly higher than MERRA-2, especially over the Northern Hemisphere's oceans; nudging reduces this difference as previously discussed. Total precipitation in the E2.1 fields has a weaker pattern correlation with MERRA-2 since the free-running model produces a split intertropical convergence zone (ITCZ) in the eastern

Pacific (a common issue in free-running GCMs, e.g., see Samanta et al., 2019); nudging the winds corrects the spatial patterns but brings the total precipitation rate out of agreement. The surface zonal wind component shows excellent agreement in their spatial patterns, although the magnitudes are greater in the E2.1 simulations relative to MERRA-2. The free-running GCM underestimates the extent of flow towards the Equator over the eastern ocean basins, which is corrected in the nudging simulation (and may be responsible for the improved ITCZ). The E2.1 simulations also lack the relatively large spatial heterogeneity seen

in surface winds over the land ice sheets.

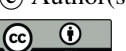

**Figure 4.** Comparison of present-day surface meteorology. Columns from left to right show annual climatological means for 2005-2014 C.E. in the MERRA-2 reanalysis, our E2.1 simulations with winds nudged to MERRA-2, and our free-running E2.1 simulations. From top to bottom, rows show surface air temperature in K, total precipitation rate in mm d$^{-1}$, the zonal component of the surface wind in m s$^{-1}$, and the meridional component of the surface wind in m s$^{-1}$. Gray dots show where the two E2.1 simulations are statistically different than their MERRA-2 counterparts with respect to interannual variability ($p$-value $< 0.05$; $n = 10$ yr). The number in the bottom left shows the global mean value for each panel. The top right number shows the pattern correlation and the number in the bottom right shows the global mean difference in the E2.1 simulations with respect to MERRA-2.

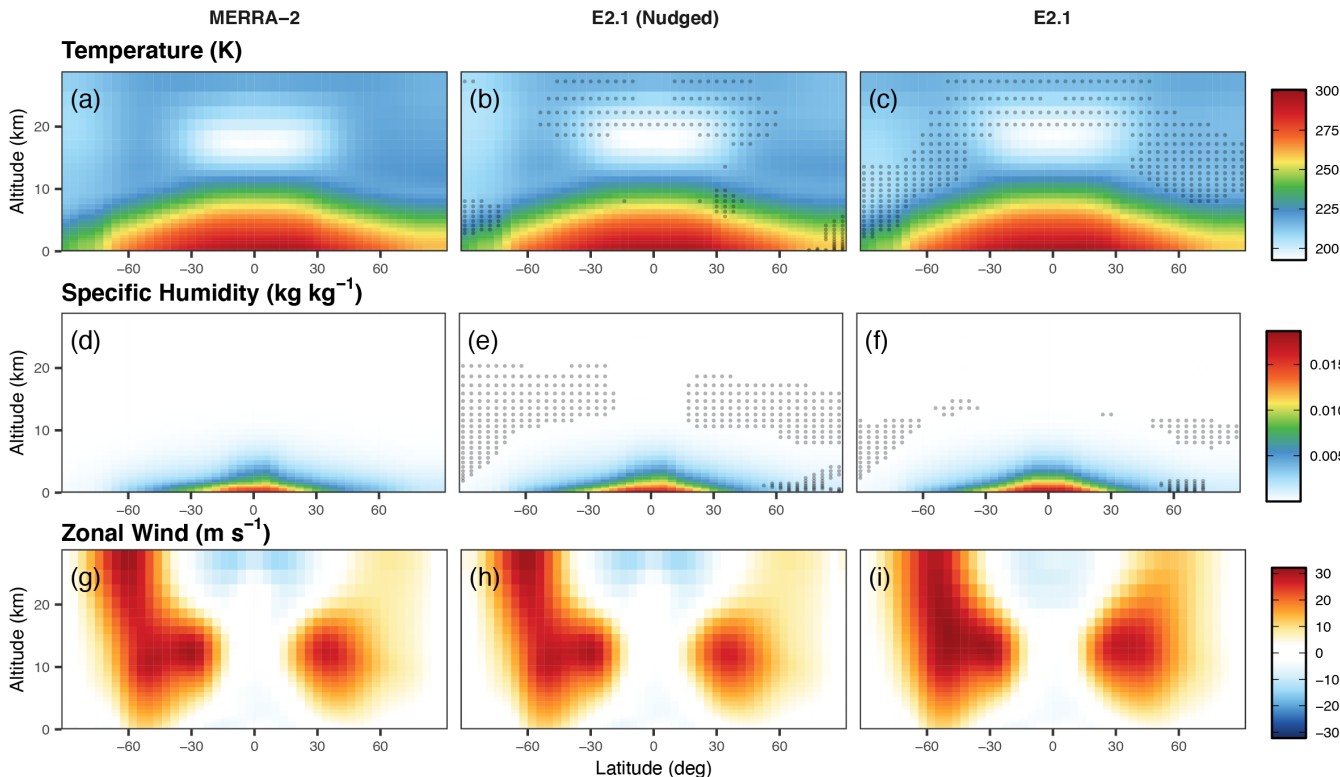

**Figure 5.** Comparison of present-day zonal meteorology. Columns from left to right show annual climatological zonal means for 2005-2014 C.E. in the MERRA-2 reanalysis, our E2.1 simulations with winds nudged to MERRA-2, and our free-running E2.1 simulations. From top to bottom, rows show air temperature in K, specific humidity in kg kg$^{-1}$, and the zonal wind component in m s$^{-1}$. Gray dots show where the two E2.1 simulations are statistically different than their MERRA-2 counterparts with respect to interannual variability ($p$-value < 0.05; $n$ = 10 yr).

Figure 5 compares key zonal mean meteorological variables generated for GCAP 2.0 with their MERRA-2 counterparts. Lower and free tropospheric air temperatures are in agreement between E2.1 and MERRA-2, but the higher tropopause leads to colder temperatures in the upper troposphere with respect to MERRA-2. Nudging the winds removes some of the temperature difference in the extratropical upper troposphere but introduces differences in the free troposphere. Specific humidity agrees between E2.1 and MERRA-2 except for a drier polar free troposphere and northern extratropical surface; nudging leads to an additional drying of the stratosphere. The zonal winds agree well between all simulations, particularly between the MERRA-2 reanalysis and the nudged simulation as to be expected.

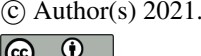



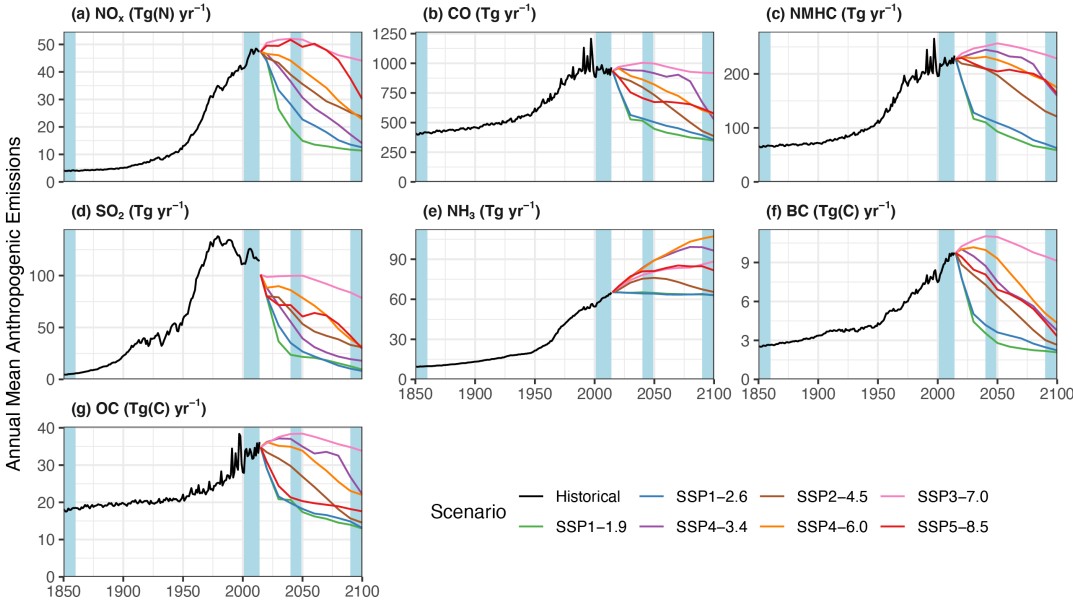

**Figure 6.** Time series of annual mean CMIP6 anthropogenic emissions for 1850-2100 C.E. for **(a)** reactive oxides of nitrogen ($NO_x \equiv NO + NO_2$) in Tg N yr$^{-1}$, **(b)** carbon monoxide (CO) in Tg yr$^{-1}$, **(c)** non-methane hydrocarbons (NMHC) in Tg yr$^{-1}$, **(d)** sulfur dioxide ($SO_2$) in Tg yr$^{-1}$, **(e)** ammonia ($NH_3$) in Tg yr$^{-1}$, **(f)** black carbon (BC) in Tg C yr$^{-1}$, and **(g)** organic carbon (OC) in Tg C yr$^{-1}$. Historical emissions for 1850-2014 C.E. from Hoesly et al. (2018) are shown as black lines. Future scenarios for 2015-2100 C.E. from Gidden et al. (2019) are shown as colored lines: SSP1-1.9 (green); SSP1-2.6 (blue); SSP4-3.4 (purple); SSP2-4.5 (brown); SSP4-6.0 (orange); SSP3-7.0 (pink); SSP5-8.5 (red). The shaded blue rectangles indicate periods for which GCAP 2.0 input meteorology is available at the time of publication.

## 5 Emissions and Boundary Conditions

This section describes the anthropogenic emission inventories and surface boundary conditions from the CMIP6 experiment that have been processed for use by GEOS-Chem, whether driven by E2.1 or MERRA-2 meteorology (Sect. 5.1). It then evaluates and compares the emission fluxes that are sensitive to meteorology between MERRA-2 and E2.1-driven GEOS-Chem simulations in the recent past (Sect. 5.2).

### 5.1 Anthropogenic Emissions and Surface Boundary Conditions

Figure 6 shows the time series of annual mean anthropogenic emissions and Figure 7 shows the time series of annual mean surface boundary conditions developed for the CMIP6 experiments and processed for use in GEOS-Chem. Emissions are used for short-lived climate forcers and air pollution precursors. Surface boundary conditions are used to prescribe long-lived species like chlorofluorocarbons that are well-mixed in the troposphere but may advect to and react within the stratosphere. The emissions and boundary conditions developed for the CMIP6 experiments include a historical reconstruction and several future sce-

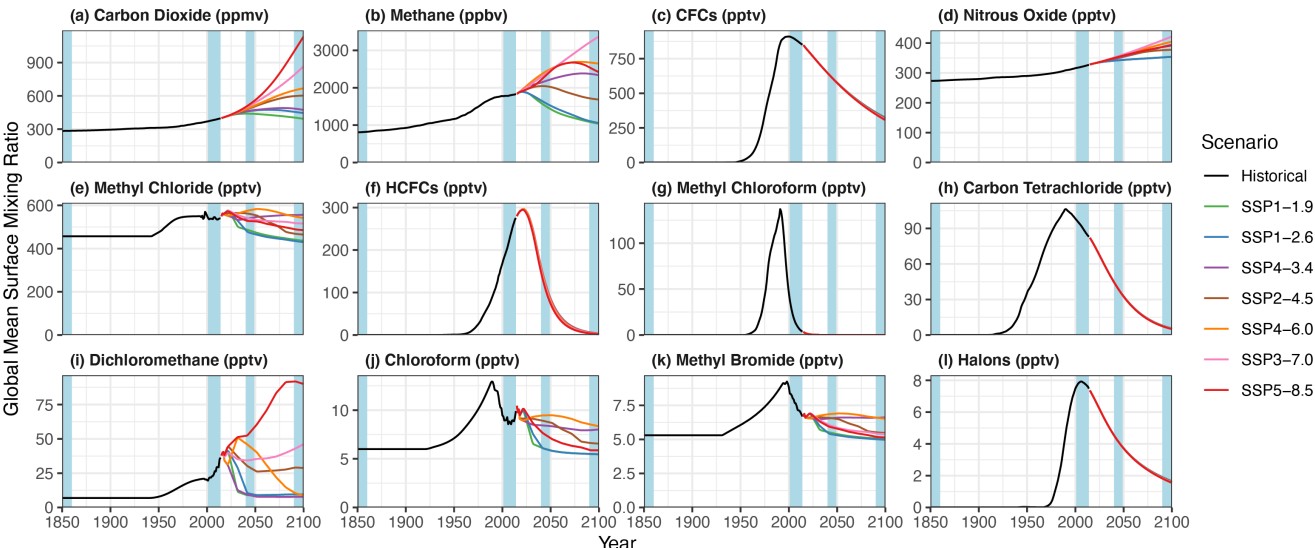

**Figure 7.** Time series of annual mean CMIP6 surface boundary conditions for 1850-2100 C.E. Individual panels show the temporal evolution of **(a)** carbon dioxide ($CO_2$) in ppmv ($\equiv \mu$mol mol$^{-1}$), **(b)** methane ($CH_4$) in ppbv ($\equiv$ nmol mol$^{-1}$), **(c)** total chlorofluorocarbons (CFCs $\equiv$ CFC11 + CFC12 + CFC113 + CFC114 + CFC115) in pptv ($\equiv$ pmol mol$^{-1}$), **(d)** nitrous oxide ($N_2O$) in pptv, **(e)** methyl chloride ($CH_3Cl$) in pptv, **(f)** total hydrochlorofluorocarbons (HCFCs $\equiv$ HCFC141b + HCFC142b + HCFC22) in pptv, **(g)** methyl chloroform ($CH_3CCl_3$) in pptv, **(h)** carbon tetrachloride ($CCl_4$) in pptv, **(i)** dichloromethane ($CH_2Cl_2$) in pptv, **(j)** chloroform ($CHCl_3$) in pptv, **(k)** methyl bromide ($CH_3Br$) in pptv, and **(l)** total halons ($\equiv$ Halon1211 + Halon1301 + Halon2402) in pptv. Historical boundary conditions are from Meinshausen et al. (2017) and future boundary conditions are described by Riahi et al. (2017). The shaded blue rectangles indicate periods for which GCAP 2.0 input meteorology is available at the time of publication.

narios with different target radiative forcings for the end of this century. They are hosted on ESGF under the input data sets for Model Intercomparison Projects (input4MIPs) project. They are available from https://esgf-node.llnl.gov/projects/input4mips. These emissions and boundary conditions have been processed for input to GEOS-Chem/GCAP 2.0. They are consistent with those that influenced the climate of the E2.1 simulations used to generate the respective GCAP 2.0 meteorologies.

### 5.1.1 Historical

Historical anthropogenic emissions in CMIP6 are from the Community Emissions Data System (CEDS, Hoesly et al., 2018). For surface emissions, we processed version 2017-05-18 of the CEDS inventory available at monthly temporal and 0.5° spatial resolution for the eight surface sectors listed in Table 2 and the 1850-2014 C.E. period. CEDS is presently the default global surface anthropogenic emissions inventory used by GEOS-Chem, although only species for the full-chemistry simulation have been processed and these emissions are overwritten for many locations by regional inventories. Here, we additionally processed the methane and $CO_2$ fluxes for those GEOS-Chem specialty simulations. All surface emissions increased exponentially over





**Table 2.** CEDS surface emission sectors.

| Sector | Description |
| --- | --- |
| agr | Agriculture (excluding crop burning) |
| ene | Energy transformation and extraction |
| ind | Industrial combustion and processes |
| rco | Residential, commercial and other |
| shp | International shipping |
| slv | Solvents |
| tra | Surface transportation (road; rail; other) |
| wst | Waste disposal and handling |

the historical period except for sulfur dioxide ($SO_2$), whose emissions peaked in the 1980s (Fig. S52). We also processed the three-dimensional CEDS aircraft emissions for input into GEOS-Chem as a CMIP6-compliant alternative to the Aviation Emissions Inventory Code (AEIC, Stettler et al., 2011) source that is the default in GEOS-Chem. We processed version 2017-08-30 of the CEDS aircraft inventory, available for NO, CO, black and organic carbon, $SO_2$ and ammonia ($NH_3$) at monthly temporal and 0.5° spatial resolution and 25 vertical levels of equal thickness from the surface to 15 km. We have vertically

re-gridded to the native 40-level E2.1 resolution and the reduced stratospheric 47-level MERRA-2 resolution. Global aircraft emissions increased mostly linearly across the historical period beginning ca. 1950 C.E. (Fig. S53).

In addition, we processed the biomass burning emissions from version 1.2 of the Biomass Burning for Model Intercomparison Projects (BB4MIPs) inventory (van Marle et al., 2017). The BB4MIPs reconstruction combines version 4 of the satellite-based Global Fire Emissions Database (GFED4; van der Werf et al., 2017, the default biomass burning inventory

in GEOS-Chem and available since 1997 C.E.) with observational proxies and model simulations from the Fire Model Intercomparison Project (FireMIP) for earlier periods. BB4MIPs provides the total mass flux per species at monthly temporal and 0.25° spatial resolution. We have re-gridded to 0.5° spatial resolution and speciated for input to GEOS-Chem using a consistent hydrocarbon speciation scheme with CEDS. Fire emissions from BB4MIPs increase slightly across the historical period. However, interannual variability greatly increases in the second half of the $20^{th}$ century (Fig. S54), driving the large

interannual variability observed in the total emissions during this period (e.g., Fig. 6b).

We have prepared version 1.2.0 of the CMIP6 surface boundary conditions derived from historical observations (Meinshausen et al., 2017) and available at monthly temporal resolution and as 0.5° latitude bands. These have been re-gridded to 0.5° global spatial resolution for input into GEOS-Chem. Carbon dioxide, methane, and nitrous oxide have monotonically increased since the preindustrial due to anthropogenic activity, whereas shorter-lived stratospheric ozone-depleting substances

peaked around the turn of the last century following their ban under the Montréal Protocol (Fig. 7).





**Table 3.** Shared Socioeconomic Pathway (SSP) narratives

| | |
|---|---|
| SSP1 | Sustainability – Taking the Green Road |
| SSP2 | Middle of the Road |
| SSP3 | Regional Rivalry – A Rocky Road |
| SSP4 | Inequality – A Road Divided |
| SSP5 | Fossil-fueled Development – Taking the Highway |

### 5.1.2 Future Scenarios

The future anthropogenic emissions and boundary conditions used by CMIP6 and processed here for GCAP 2.0 are summarized by Riahi et al. (2017). In brief, the so-named "Shared Socioeconomic Pathways" (SSPs) are determined from integrated assessment modeling (IAM) of five future societal narratives that may be followed to limit future warming to a target radiative forcing. The nomenclature for characterizing the SSP scenarios is SSP$x$-$y.z$ (or SSP$xyz$), where $x$ is the number of the future narrative (1 to 5; Table 3), and $y.z$ represents the target radiative forcing in W m$^{-2}$ at the end of the $21^{st}$-century ranging from 1.9 (extreme mitigation; low warming) to 8.5 (low mitigation; extreme warming). The names of the five narratives are listed in Table 3, and each employs different assumptions about how global society would achieve the target radiative forcing. SSP1 assumes low challenges to mitigation and adaptation (van Vuuren et al., 2017), SSP2 assumes medium challenges to mitigation and adaptation (Fricko et al., 2017), SSP3 assumes high challenges to mitigation and adaptation (Fujimori et al., 2017), SSP4 assumes low challenges to mitigation and high challenges to adaptation (Calvin et al., 2017), and SSP5 assumes high challenges to mitigation and low challenges to adaptation (Kriegler et al., 2017).

For GCAP 2.0, we focus on seven scenarios corresponding to Tiers 1 and 2 of the ScenarioMIP experiment: SSP1-1.9, SSP1-2.6, SSP4-3.4, SSP2-4.5, SSP4-6.0, SSP3-7.0, and SSP5-8.5 (O'Neill et al., 2016). Assumptions about future population growth, urbanization, gross domestic production (GDP), energy, land-use, and air pollution trends of the SSP IAM scenarios are described in a series of manuscripts (Crespo Cuaresma, 2017; Dellink et al., 2017; Jiang and O'Neill, 2017; Leimbach et al., 2017; Samir and Lutz, 2017; Bauer et al., 2017; Popp et al., 2017; Rao et al., 2017). Version 1.1 of these scenarios (Gidden et al., 2019) were obtained from input4MIPs on ESGF and available at 0.5° spatial and monthly resolution for 2015 C.E. and then every 10 years beginning with 2020 C.E. and ending with 2100 C.E. They were processed for input to GEOS-Chem in the same way as the historical emissions and boundary conditions. Linear interpolation was used to develop individual yearly emissions between the available decadal values. In addition to the sectors of Table 2, future $CO_2$ emissions include a "neg" sector that considers negative emission (i.e., carbon capture).

From the perspective of the simulated climate, the forcing that dominates the end-of-the-century response is the $CO_2$ abundance. Therefore, $CO_2$ is the only gas that monotonically increases with future forcing target values (Fig. 7a). The trajectories of the remainder of the well-mixed greenhouse gases, stratospheric ozone-depleting substances, short-lived climate forcers, and air-pollution precursors vary between the SSP scenarios and target forcings (Figs. 6 and 7b-l). For example, methane in 2100 C.E. is highest in the SSP3-7.0 scenario, not SSP5-8.5 (Fig. 7b). Because ammonia is primarily produced from agriculture





and the world population will continue to grow, it is the only emission expected to remain constant or grow into the future. Otherwise, the narrative assumptions strongly influence the global and regional emission changes. Furthermore, we note that

the historical emissions inventories have large amounts of interannual variability (primarily due to biomass burning), whereas the SSP scenarios have very low interannual variability. This highlights the necessity for CTM studies such as those that may be accomplished by GCAP 2.0 that can explore the impact of a wider range of future emission trajectories on air quality, short-lived climate forces, and stratospheric ozone in future warmer climates whose meteorology is primarily driven by changes in $CO_2$. It also highlights the necessity for performing simulations long enough to establish robust statistics for chemistry-climate

interactions (e.g., Garcia-Menendez et al., 2017).

### 5.2 Natural Emissions

Figure 8 shows the climatological mean emission fluxes for key species whose emissions are sensitive to meteorology. Figures S55-S60 in the supplementary materials provide seasonal detail for each species.

Emission fluxes sensitive to meteorology and thereby grid resolution began to be pre-processed offline at high spatial res-

olution in version 12.4.0 of GEOS-Chem. This was to facilitate the calculation of consistent emissions between the various cubed-sphere geometries of the GCHP variant of the model (Eastham et al., 2018, see Sect. 3.0.2), although the option to calculate these emissions online was maintained through online extensions in the Harmonized EMissions COmponent (HEMCO) processing code (Keller et al., 2014). The default behavior for GCAP 2.0 is to use online calculations to respond to the underlying climate. Users who wish to run GCHP with GCAP 2.0 may quickly pre-process offline natural emission fluxes using the

"standalone" version of HEMCO with a HEMCO_Config.rc file from a GCClassic version of GCAP 2.0. Here we compare the online emission fluxes in our MERRA-2-driven simulation to those of our E2.1-driven simulations.

Panels a-c of Fig. 8 show the spatial distribution of isoprene from terrestrial plants. Emissions of non-methane hydrocarbons (NMHCs) from terrestrial plants follow version 2.1 of the Model of Emissions from Gases and Aerosols from Nature (MEGAN), which responds positively to changes in diffuse photosynthetically active radiation (PAR), recent surface air tem-

perature, and soil root wetness (Guenther et al., 2012). There is an option for emissions to respond to $CO_2$ abundance as well (Tai et al., 2013). Because E2.1 has a greater proportion of PAR present as diffuse radiation, isoprene emissions in E2.1-driven simulations are about 40 % higher than in the MERRA-2-driven simulation.

Panels d-f of Fig. 8 show a very tight agreement in the spatial pattern and magnitude of the flux of dimethylsulfide (DMS; $(CH_3)_2S$) produced by marine phytoplankton. Emissions of NMHCs from marine environments are represented as the prod-

uct of prescribed seawater concentration distributions and sea-to-air transfer velocities calculated via the parameterization of Nightingale et al. (2000a, b). The latter respond to sea-surface temperatures and surface wind velocities.

Panels g-i of Fig. 8 compare the source of mineral dust between the simulations. We use the Dust Entrainment And Deposition (DEAD) scheme for mineral dust evasion (Zender et al., 2003), which responds to changes in surface friction velocity ($u^*$), roughness height, snow/ice cover and depth, soil wetness, pressure, specific humidity and temperature. Dust mobilization was

found in our tests to be extremely sensitive to the meteorology product used, with poor spatial correlation ($R \leq 0.26$) and with each meteorology product yielding a different order of magnitude in its global total. Therefore, we have determined respective







**Figure 8.** Annual mean spatial distribution of meteorology-dependent emission fluxes for 2005-2014 C.E. Each column from left to right shows emission fluxes calculated using: MERRA-2 meteorology, E2.1 meteorology nudged to MERRA-2, and the free-running E2.1 meteorology, respectively. Each row from top to bottom shows emission fluxes for: **(a-c)** isoprene (2-methyl-1,3-butadiene; $CH_2=C(CH_3)CH=CH_2$) from terrestrial plants in $10^{-9}$ kg m$^{-2}$ s$^{-1}$, **(d-f)** dimethylsulfide (($CH_3)_2$S; DMS) from marine organisms, **(g-i)** aeolian mineral dust, **(j-l)** aeolian sea salt, **(m-o)** the vertically integrated source of NO from lightning, and **(p-r)** NO from soil microbial activity, respectively. Gray dots indicate locations where the two E2.1-driven simulations show statistically significant differences ($p$-value $< 0.05$; $n = 10$ years) with respect to the MERRA-2-driven simulation. The value in the lower left of each panel gives the globally integrated source in **(a-l)** Tg yr$^{-1}$ or **(m-r)** Tg N yr$^{-1}$. The number in the lower (upper) right of each panel gives the total difference (pattern correlation) of the E2.1-driven simulations with respect to their respective MERRA-2-driven values.





scaling factors for the E2.1 simulations that bring the present-day global total into agreement with the MERRA-2-driven value. These are included by default in the GCAP 2.0 run directories for the DEAD dust scheme.

Panels j-l of Fig. 8 show the source of sea salt aerosol. The sea-salt mobilization scheme is described by Jaeglé et al. (2011)
and responds to sea-surface temperature, surface wind velocity, and the fraction of sea-ice coverage. There is an excellent agreement between each source's spatial distributions, although the stronger surface winds in E2.1 lead to 15-20 % higher emissions in the E2.1 simulations, especially over the Southern Ocean.

Panels m-o of Fig. 8 show the column-integrated source of NO from lightning. In MERRA-2, lightning flash densities (flashes $km^{-2}$ $s^{-1}$) are pre-calculated offline from MERRA-2 convective cloud depths at 0.5° latitude by 0.625° longitude
spatial and 3-h temporal resolution. These densities are then input as a meteorological parameter to GEOS-Chem, from which vertical profiles of NO production are determined following Murray et al. (2012). For the GCAP 2.0 meteorology, flash rates are calculated online in the E2.1 moist convection code following the description in Kelley et al. (2020) and archived at E2.1 native spatial and hourly temporal resolution for input as a meteorological parameter to GEOS-Chem (see Sect. 4). Both lightning flash density calculations are ultimately based on the same cloud-top height scheme of Price and Rind (1992) and global
mean lightning flash rates are tuned to climatology in the recent past (Cecil et al., 2014). However, because the spatial and seasonal climatology in the MERRA-2-driven simulations is constrained by satellite observations (Murray et al., 2012), which is not appropriate for a free-running GCM, the spatial patterns differ between the simulations. Therefore, E2.1 overestimates the fraction of lightning in the tropics with respect to the extratropics, and puts too much lightning over South America and Oceania and not enough over Africa. It is also worth emphasizing that we do not know how lightning has changed since the
preindustrial or will change in a warming world (e.g., Williams, 2005; Price, 2013; Murray, 2016, 2018; Finney et al., 2018).

Panels p-r of Fig. 8 show the source of NO from soil microbial activity. The parameterization is described by Hudman et al. (2012) and responds to surface air temperature, wind speed, soil wetness, cloud fraction, downwelling shortwave radiation, and snow/ice cover. To a lesser degree, lightning can also influence the soil NO source through its impact on nitrate deposition to the soils. The spatial correlation is excellent between the different sources, although the E2.1 magnitude is higher by about
370 40 %.

Lastly, we note the important and variable geologic source of $SO_2$. Volcanic emissions of $SO_2$ in GEOS-Chem are normally prescribed from the Aerosol Comparisons between Observations and Models (AeroCom) point-source inventory (Carn et al., 2015), with data available since 1978 C.E. The CMIP6 experiment did not provide historical or future emission fluxes for volcanism. Instead, input4MIPs provided time series of stratospheric aerosol surface area densities and effective radii with
which to force the GCMs. Therefore, when users generate a GCAP 2.0 run directory, they are given the option to select a fixed historical AeroCom year from which to prescribe their volcanic emissions.





## 6 Model Evaluation

This section evaluates the performance of GCAP 2.0 driven by E.21 versus MERRA-2 meteorology for the recent past through comparison with observations. We first evaluate model physics and transport using the "TransportTracers" variant of GEOS-
Chem (Sect. 6.1). We then evaluate the standard full chemistry mechanism (Sect. 6.2).

All simulations were performed at $4°$ latitude by $5°$ longitude horizontal resolution for the period 2001-2014 C.E., with meteorology respectively prescribed from MERRA-2, E2.1 nudged to MERRA-2, and the free-running E2.1 simulation. All simulations used version 12.9.3 of GEOS-Chem (doi:10.5281/zenodo.3974569) with modifications as described throughout the manuscript. The E2.1 meteorology fields were re-gridded from their native resolution upon input to GEOS-Chem by FlexGrid.
The E2.1 simulations used the native 40-layer resolution and the MERRA-2 simulations used the 47-layer reduced-stratospheric resolution (see Fig. 1). Identical initial conditions were re-gridded to each model's respective vertical resolution. The first four years of each simulation were discarded as initialization, with the remaining ten years used for evaluation and statistics. All prescribed emissions were identical between each simulation.

### 6.1 Transport

Model transport and physical processes may be evaluated against observations using the "TransportTracers" variant of GEOS-Chem. We focus on four tracers of particular utility: sulfur hexafluoride ($SF_6$), radon-222 ($^{222}Rn$), lead-210 ($^{210}Pb$), and beryllium-7 ($^7Be$).

Sulfur hexafluoride is a trace gas of anthropogenic origin that is chemically and physically inert on human time scales (lifetime of 3200 yr). It is primarily emitted at the surface in the northern hemisphere (Maiss and Brenninkmeijer, 1998). Its
meridional gradient may be used to test the rate of inter-hemispheric mixing (Rigby et al., 2010; Hall et al., 2011) and its vertical gradients may be used to infer the age of air in the stratosphere (Waugh and Hall, 2002; Waugh, 2009). Its meridional gradients may also be used to infer the tropospheric age of air (Waugh et al., 2013). Here we use emissions from version 4.2 of the Emissions Database for Global Atmospheric Research (EDGAR), available at $0.1°$ global resolution for 1970-2008 (https://doi.org/10.2904/EDGARv4.2).

Terrigenic $^{222}Rn$ is an inert, insoluble, short-lived (half-life 3.8 d) noble gas produced from the slow decay of $^{226}Ra$ (half-life 1600 yr) found in uranium ores. Its evasion from surface soils is relatively uniform and constant and is as described by Jacob et al. (1997). Its insolubility and time scale of decay make it a useful tracer for diagnosing quick vertical mixing within atmospheric models from boundary layer processes and moist convection (e.g., Allen et al., 1996; Brost and Chatfield, 1989; Considine et al., 2005; Feichter and Crutzen, 1990; Hauglustaine et al., 2004; Jacob and Prather, 1990; Jacob et al., 1997;
Lambert et al., 1982; Mahowald et al., 1995; Stockwell et al., 1998).

Radiogenic $^{210}Pb$ is the chemically-inert decay product of $^{222}Rn$. It is readily taken up by sub-micron aerosol particles and subsequently removed from the atmosphere by deposition or decay (Bondietti et al., 1988; Maenhaut et al., 1979; Preiss et al., 1996; Sanak et al., 1981). Because of its relatively long lifetime (half-life 22.2 yr), nearly all $^{210}Pb$ is removed via deposition.





**Table 4.** Radionuclide budgets for 2005-2014 CE.

|  |  | MERRA-2 | E2.1 (Nudged) | E2.1 |
|---|---|---|---|---|
| **Radon-222** |  |  |  |  |
| Global Burden, g |  | 188 | 188 | 189 |
| Troposphere |  | 187 (99.5 %) | 188 (99.7 %) | 189 (99.8 %) |
| Stratosphere |  | 1 (0.5 %) | 1 (0.3 %) | 0 (0.2 %) |
| Sources, g d$^{-1}$ |  | 34 | 34 | 34 |
| Sinks, g d$^{-1}$ |  | 34 | 34 | 34 |
| Tropospheric residence time, d |  | 0.5 | 0.5 | 0.5 |
| **Lead-210** |  |  |  |  |
| Global Burden, g |  | 321 | 322 | 316 |
| Troposphere |  | 269 (83.8 %) | 276 (85.8 %) | 281 (89.0 %) |
| Stratosphere |  | 52 (16.2 %) | 46 (14.2 %) | 35 (11.0 %) |
| Sources, g d$^{-1}$ |  | 32 | 32 | 32 |
| Sinks, g d$^{-1}$ |  | 32 | 32 | 32 |
| Radioactive Decay: | Troposphere | 0 (0.1 %) | 0 (0.1 %) | 0 (0.1 %) |
|  | Stratosphere | 0 (0.0 %) | 0 (0.0 %) | 0 (0.0 %) |
| Dry Deposition |  | 4 (12.7 %) | 3 (9.4 %) | 3 (9.4 %) |
| Wet Deposition: | Stratiform | 18 (56.3 %) | 1 (3.1 %) | 1 (3.1 %) |
|  | Convective | 10 (30.9 %) | 28 (87.5 %) | 28 (87.5 %) |
| Tropospheric residence time, d |  | 8.3 | 8.8 | 8.8 |
| **Beryllium-7** |  |  |  |  |
| Global Burden, g |  | 16 | 15 | 15 |
| Troposphere |  | 3 (22.3 %) | 4 (26.4 %) | 4 (30.5 %) |
| Stratosphere |  | 12 (77.7 %) | 11 (73.6 %) | 10 (69.5 %) |
| Sources, g d$^{-1}$ |  | 0.33 | 0.34 | 0.34 |
| Cosmogenic: | Troposphere | 0.12 (37.1 %) | 0.14 (42.8 %) | 0.15 (46.1 %) |
|  | Stratosphere | 0.21 (62.9 %) | 0.19 (57.2 %) | 0.18 (53.9 %) |
| Sinks, g d$^{-1}$ |  | 0.33 | 0.34 | 0.34 |
| Radioactive Decay: | Troposphere | 0.05 (13.6 %) | 0.05 (15.2 %) | 0.06 (17.2 %) |
|  | Stratosphere | 0.16 (47.2 %) | 0.14 (42.6 %) | 0.13 (39.2 %) |
| Dry Deposition |  | 0.01 (3.6 %) | 0.01 (3.7 %) | 0.01 (3.8 %) |
| Wet Deposition: | Stratiform | 0.09 (25.7 %) | 0.01 (1.9 %) | 0.01 (2.2 %) |
|  | Convective | 0.03 (9.9 %) | 0.12 (36.6 %) | 0.13 (37.5 %) |
| Tropospheric residence time, d |  | 19.7 | 20.4 | 21.7 |





As its source from $^{222}$Rn is relatively well known, and there is a global and long-term surface deposition flux inventory (Preiss
et al., 1996), it is the standard test for model deposition.

Cosmogenic $^7$Be is produced by cosmic-ray spallation of $N_2$ and $O_2$, predominantly in the polar upper troposphere and lower stratosphere (Lal et al., 1958). The source of $^7$Be is updated for this work to use the parameterization of Usoskin and Kovaltsov (2008). Mean solar activity is assumed (solar modulation potential $\Phi = 670$ MV), leading to an average production rate of 0.065 atoms cm$^{-2}$ s$^{-1}$; about 60 % in the stratosphere and 40 % in the troposphere. Like $^{210}$Pb, $^7$Be is rapidly taken up by
sub-micron aerosol particles (Bondietti et al., 1988; Maenhaut et al., 1979; Papastefanou, 2009; Papastefanou and Ioannidou, 1996; Sanak et al., 1981). It is subsequently transported until removal by deposition or radioactive decay (half-life 53.3 d). $^7$Be has been used to constrain vertical transport, wet deposition fluxes, and stratosphere-troposphere exchange in models (e.g., Allen et al., 2003; Brost et al., 1991; Koch et al., 1996; Liu et al., 2001, 2016; Barrett et al., 2012).

Table 4 gives the atmospheric budget of the three radionuclides driven by the three meteorological products.

**6.1.1 Horizontal Mixing**

Figure 9 shows observed meridional and vertical gradients of SF$_6$ with respect to Cape Matatula, American Samoa (SMO) in the remote tropical southern Pacific. The observations are version 2.1.1 of the NOAA Carbon Cycle Group SF$_6$ ObsPack (doi:10.25925/20180817) and represent a mixture of surface *in situ*, flask, tower, and aircraft sources from 2005-2014 C.E. Observations were aggregated at model spatial and monthly temporal resolution, compared to that month's SMO value, from
which zonal climatologies were determined. Also shown is the value of each simulation sampled and processed as in the observations. In all simulations, GEOS-Chem underestimates the cross-equatorial meridional gradient by 17-26 %, an improvement over GEOS-Chem driven by earlier meteorology products (see supplementary materials of Murray et al., 2014). However, this suggests that the inter-hemispheric mixing rate in the model is too fast and/or that the EDGAR inventory underestimates the emission growth rate in the northern hemisphere relative to the southern hemisphere. Meanwhile, meridional mixing rates in the
Southern Hemisphere are consistent with the observations in all simulations. The E2.1-simulation slightly better matches the cross-equatorial gradient than the MERRA-2 and nudged simulations, but E2.1 greatly underestimates the lower stratospheric gradient (see Sect. 6.1.3).

**6.1.2 Vertical Mixing - Troposphere**

We assess vertical mixing within the troposphere using vertical profiles of $^{222}$Rn and the ratio of $^7$Be to $^{210}$Pb in surface air.
Figure 10 shows simulated climatological $^{222}$Rn profiles sampled at the month and location of the available observations, also plotted. Observations are scarce and available only at northern extratropical continental locations (Bradley and Pearson, 1970; Nazarov et al., 1970; Wilkening, 1970; Moore et al., 1973; Kritz et al., 1998). In an overly convective atmosphere, the vertical gradient of $^{222}$Rn would disappear. There is a slight overestimate within the boundary layer and underestimate above in all of our simulations, implying a small underestimate in boundary layer ventilation. Our results are comparable to or better
than other atmospheric models (e.g., see Fig. 5 of Considine et al., 2005).

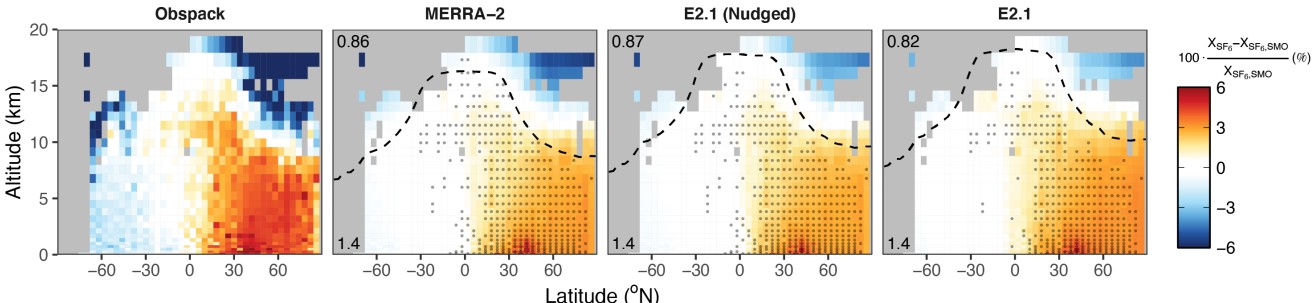

**Figure 9.** Lower atmospheric gradients of $SF_6$ shown as zonal mean percent difference with respect to Cape Matatula, American Samoa (SMO; $14.2°$S, $170.6°$W; 42 m.a.s.l.) for 2005-2014 C.E. The left panel shows observed values from version 2.1.1 of the NOAA $SF_6$ ObsPack aggregated at E2.1 vertical resolution. The right three panels show the values from the model driven by the three meteorology products sampled at each observation month and location. Gray dots indicate locations where the simulated gradient is statistically different than the observations with respect to interannual variability ($p$-value $< 0.05$; $n = 10$ yr). The number in each model panel's top left shows the pattern correlation ($R$) of that simulation to the observations. The lower left number shows the mean absolute percent bias of each simulation relative to the observations. The dashed black line shows each simulation's climatological zonal mean tropopause height.

Figure 11 shows the annual mean surface mixing ratios of $^7$Be, $^{210}$Pb and their ratio in our simulations. Since $^7$Be is produced in the upper troposphere/lower stratosphere and $^{210}$Pb is produced near the surface, and because the ratio of $^7$Be to $^{210}$Pb is unaffected by deposition, the ratio serves as a useful measure for tropospheric vertical mixing (Koch et al., 1996). A persistent high bias would indicate excessive downward transport and/or insufficient upward transport, assuming no bias in either source. The left column shows the climatological long-term data from the surface monitoring stations of the DOE Environmental Measurements Laboratory (EML) Surface Air Sampling Program (SASP) (obtained Jan 2021 from http://www.wipp.energy.gov/namp/emllegacy/databases.htm). SASP recorded the spatial and temporal distribution of various radionuclides in surface ambient air from 1957 until 1999 C.E., including $^7$Be and $^{210}$Pb. For comparison with our simulations, we select data from periods of average solar activity (solar modulation potential $\Phi = 670 \pm 50$ MV from the Usoskin et al. (2005) reconstruction). All simulations show only minor biases in surface abundance in either $^7$Be and $^{210}$Pb. The $^7$Be to $^{210}$Pb ratio shows tight agreement between the different model simulations, with the E2.1 simulation slightly outperforming the MERRA-2-driven simulation relative to the observations.

### 6.1.3 Vertical Mixing - Stratosphere

Figure 12 shows the simulated zonal climatology of the age of air in the stratosphere, which is defined as the mean time since an air mass at a given location was last in the troposphere (Hall and Waugh, 2000). Age of air increases away from the equatorial tropopause where most tropospheric air enters the stratosphere (Holton et al., 1995). We determine age of air by using $SF_6$ as a chronological tracer (e.g., Waugh and Hall, 2002) to determine the average temporal lag between a mixing ratio at a given location in the stratosphere relative to the tropical tropopause for the period 2005-2014 C.E.





**Figure 10.** Mean observed vertical profile of $^{222}$Rn compared to the model sampled at month and location of observations. The units are mBq per standard cubic meter at $0°$C and 1 atm, equivalent to a linear transformation of the molar mixing ratio (5.637 mBq SCM$^{-1}$ = 1.0 $\times 10^{-22}$ mol $^{222}$Rn / mol air).





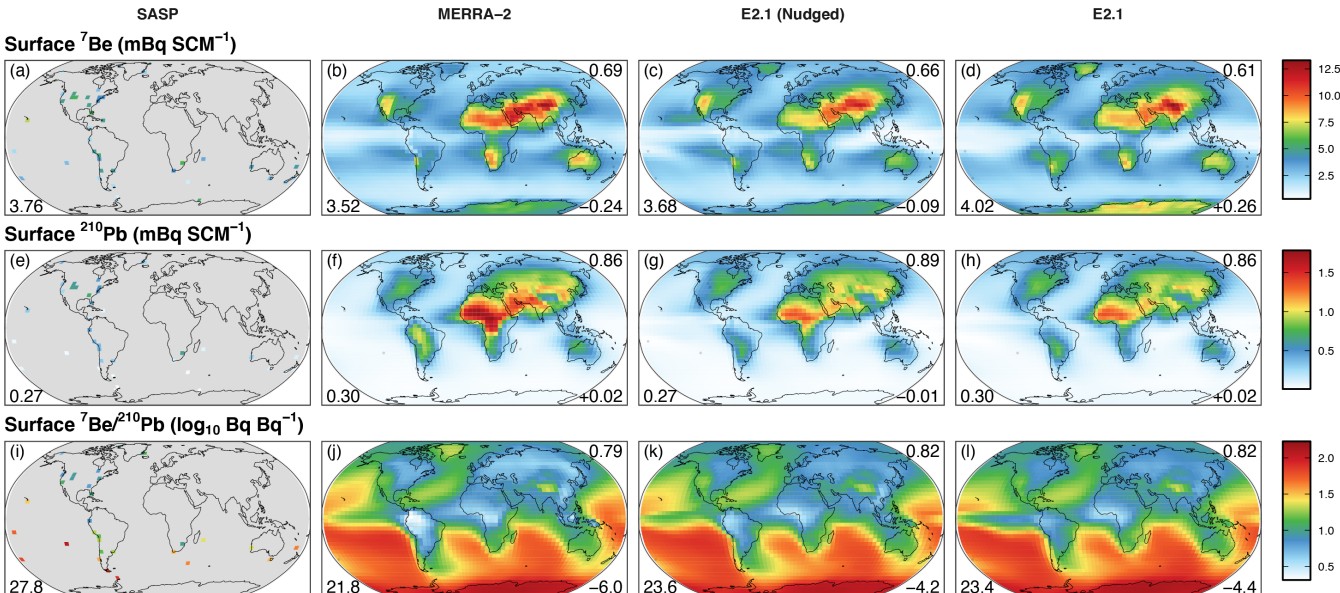

**Figure 11.** Surface mixing ratios of $^7$Be (top row; in mBq SCM$^{-1}$; 4.05 mBq SCM$^{-1}$ = 1.0 $\times 10^{-21}$ mol $^7$Be / mol air), $^{210}$Pb (middle row; in mBq SCM$^{-1}$; 2.66 mBq SCM$^{-1}$ = 1.0 $\times 10^{-19}$ mol $^{210}$Pb / mol air), and the log$_{10}$-transformed ratio of the $^7$Be to $^{210}$Pb activities in surface air (bottom row; unitless). The left column shows long-term mean observations from the DOE Surface Air Sampling Program (SASP) program for 1969-1999 C.E. The $^7$Be observations have been selected for periods of average solar activity ($\Phi = 670 \pm 50$ MV from Usoskin et al., 2005). The right three columns show from left to right: values simulated by GEOS-Chem driven by meteorology archived for 2005-2014 C.E. from MERRA-2, E2.1 nudged to MERRA-2, and the free-running E2.1. The number in each panel's lower left shows the mean value of the observations or models sampled at the observed locations. The top right number shows the pattern correlation ($R$) of the simulated values with the respective observations. The lower right number shows the mean bias of the simulated values with respect to the observations.

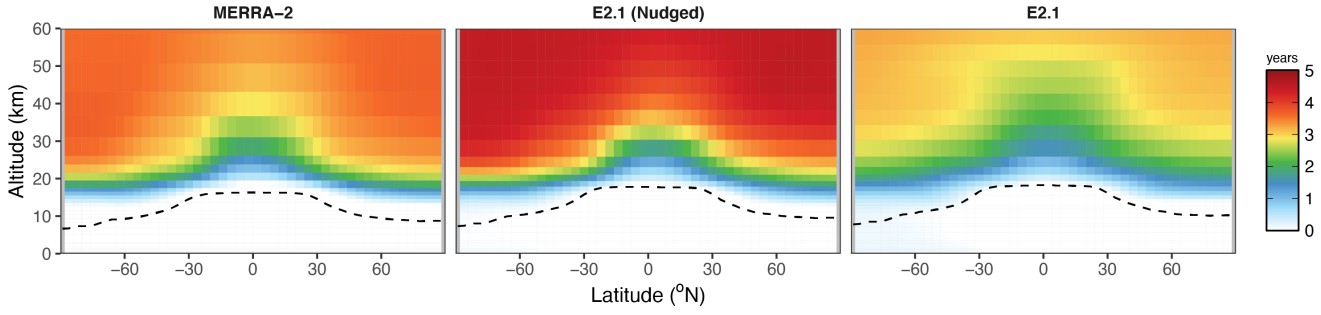

**Figure 12.** Average age of air in the stratosphere in GEOS-Chem simulations driven by MERRA-2, E2.1 nudged to MERRA-2, and E2.1, using the temporal lag in the simulated 2005-2014 C.E. time series of SF$_6$ relative to the tropical tropopause as a chronological tracer. The dashed black line shows the simulated zonal mean tropopause altitude.

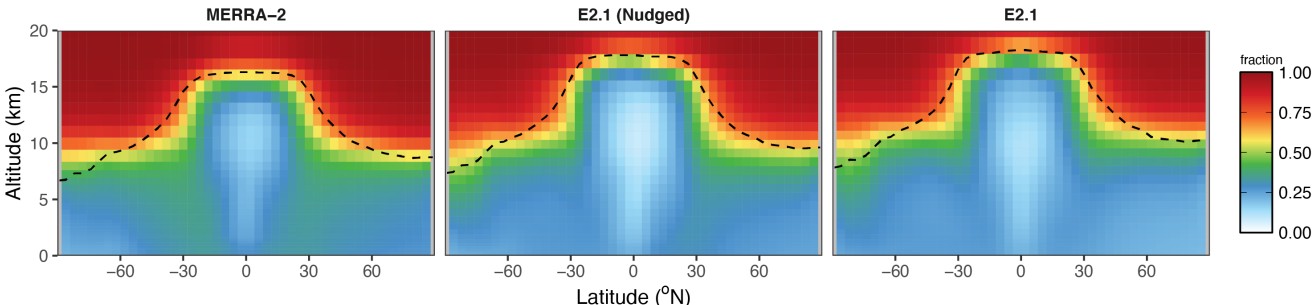

**Figure 13.** Zonal mean fraction of [7]Be that was produced in the stratosphere in GEOS-Chem driven by MERRA-2, E2.1 nudged to MERRA-2 and E2.1. Zonal mean tropopause height in each simulation is shown as a dashed line.

Models traditionally underestimate the stratospheric age of air implied by observations, which increases up to 7 years by
35 km in the poles (Waugh and Hall, 2002). All three of our simulations, including the MERRA-2 reanalysis, underestimate
the age of air. E2.1 has the youngest air with values consistent with those determined online within the GCM and reported
by Orbe et al. (2020b). E2.1 nudged to MERRA-2 has the oldest air but is still only about two thirds of what observational
constraints suggest they should be and remain difficult to interpret given the challenges of nudged simulations (e.g., Orbe et al.,
2020a). The young age in E2.1 results from too strong an ascent in the tropical pipe and a relatively leaky lower branch of the
Brewer-Dobson circulation (e.g., Ray et al., 2010); updates to version E2.2 of the GISS GCM greatly improve the stratospheric
circulation (Orbe et al., 2020b) and will be included in future GCAP 2.0 meteorology products and scenarios.

### 6.1.4 Stratosphere-Troposphere Exchange

Beryllium-7 has often been used as a tracer of downward transport from the stratosphere (Dibb et al., 1992, 1994; Husain et al.,
1977; Rehfeld and Heimann, 1995; Sanak et al., 1985; Viezee and Singh, 1980) and as an indicator of STE performance within
global atmospheric models (Allen et al., 2003; Liu et al., 2001, 2016; Barrett et al., 2012; Murray et al., 2014).

Figure 13 shows the annual zonal fraction of [7]Be of stratospheric origin in each simulation. Using E2.1 (MERRA-2) mete-
orology, we find that 23 % (30 %) of annual average surface [7]Be abundance from 38-51°N is of stratospheric origin, slightly
lower (higher) than the observational constraint of 25 % reported by Dutkiewicz and Husain (1985). This is a dramatic improve-
ment over the earlier GCAP / ICECAP studies in which the stratospheric downwelling source in the E2-driven simulations was
greatly overestimated (Murray et al., 2014) and was also seen in the NASA Global Modeling Initiative (GMI) CTM driven
by GISS Model II' meteorology (Liu et al., 2016). This possibly reflects improvements in downward mass flux in the GCM
associated with the increase in vertical resolution (Fig. 1) and updates to the GEOS-Chem dynamical core code that occurred
since the original GCAP/ICECAP.

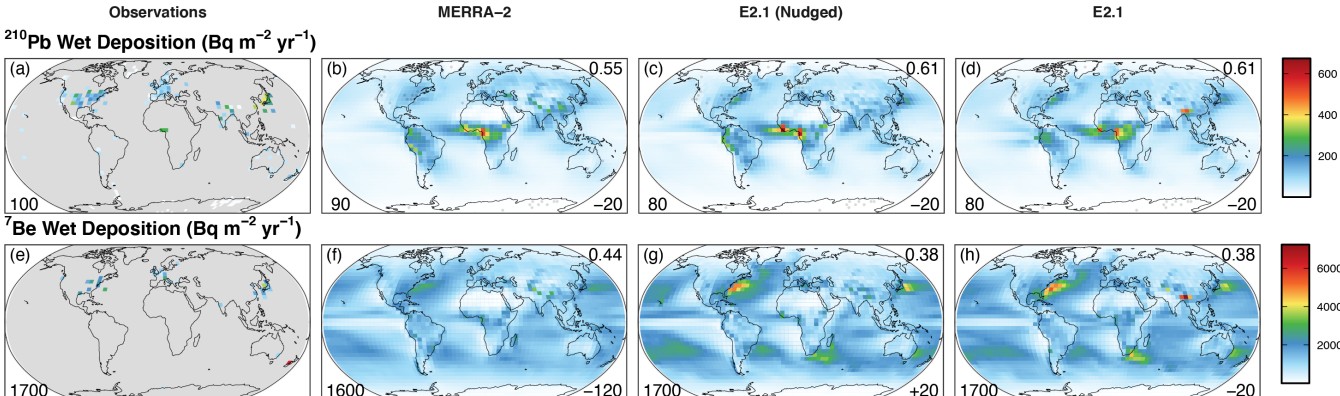

**Figure 14.** Radionuclide wet deposition fluxes. Annual average wet deposition flux in mBq m$^{-2}$ s$^{-1}$ for $^{210}$Pb (top row) and $^7$Be (bottom row). The left column shows observed values aggregated to the model resolution for $^{210}$Pb and $^7$Be. The right three columns show from left to right: values simulated by GEOS-Chem driven by meteorology archived for 2005-2014 C.E. from MERRA-2, E2.1 nudged to MERRA-2, and free-running E2.1. The number in each panel's lower left shows the mean value of the observations or models sampled at the observed locations. The top right number shows the pattern correlation ($R$) of the simulated values with the respective observations. The lower right number shows the mean bias of the simulated values with respect to the observations.

### 6.1.5 Deposition

Figure 14 (top row) compares the simulated wet deposition flux of $^{210}$Pb in each simulation to the observational values reported by Preiss et al. (1996) aggregated to model resolution. Wet deposition of $^{210}$Pb in all simulations is biased low by ∼20 % in all simulations, consistent with earlier versions of GEOS-Chem (Murray et al., 2014). These results imply a low bias in the $^{210}$Pb source (and therefore $^{222}$Rn emission) or a high bias in the dry deposition flux in all simulations.

Figure 14 (bottom row) compares the simulated wet deposition fluxes of $^7$Be to the few observations that exist (Baskaran
et al., 1993; Bleichrodt, 1978; Brown et al., 1989; Dibb, 1989; Du et al., 2008; Harvey and Matthews, 1989; Hasebe et al., 1981; Hirose et al., 2004; Igarashi et al., 1998; Narazaki and Fujitaka, 2010; Nijampurkar and Rao, 1993; Olsen et al., 1985; Papastefanou et al., 1995; Schuler et al., 1991; Turekian et al., 1983; Wallbrink and Murray, 1994, and references therein). The wet deposition flux is increased in the E2.1 simulation with respect to the MERRA-2 simulation, particularly over the mid-latitude oceans, and is the closest simulation to matching the observations.

We find a tropospheric residence time for $^{210}$Pb-containing aerosols against deposition of 8.3 d in MERRA-2 versus 8.8 d in the E2.1 simulations (Table 4), an improvement in consistency with respect to GCAP / ICECAP, and within the range of previous estimates of 6.5-12.5 d (Turekian et al., 1977; Lambert et al., 1982; Balkanski et al., 1993; Koch et al., 1996; Guelle et al., 1998b, a; Liu et al., 2001). We find a similar relative increase in the lifetime of $^7$Be-containing aerosols of 19.7 d versus 21.7 d, also consistent with earlier findings of 23 d (Koch et al., 1996) and 21 d (Liu et al., 2001). Whereas wet
deposition in MERRA-2-driven simulations is primarily due to stratiform clouds, it is primarily due to convective clouds in





**Table 5.** Lifetime against oxidation by tropospheric OH (yr).

|  | Observations[a] | MERRA-2[b] | E2.1 (Nudged)[b] | E2.1[b] |
|---|---|---|---|---|
| $CH_3CCl_3$ | $6.0^{+0.5}_{-0.4}$ | $5.3 \pm 0.0$ | $5.6 \pm 0.1$ | $5.6 \pm 0.1$ |
| $CH_4$ | $11.2 \pm 1.3$ | $9.0 \pm 0.1$ | $9.3 \pm 0.2$ | $9.3 \pm 0.2$ |

[a] Prinn et al. (2005) for $CH_3CCl_3$ and Prather et al. (2012) for $CH_4$

[b] 2005-2014 C.E. annual mean and standard deviation

E2.1-driven simulations, consistent with the meteorological fields (see Sect. 4) and leading to the longer residence times in the E2.1 simulations.

## 6.2 Chemistry

Next, we evaluate the "standard" full chemistry simulation of version 12.9.3 of GEOS-Chem driven by the three meteorology
products for 2005-2014 C.E. The standard mechanism contains a unified chemical mechanism of ozone-$NO_x$-hydrocarbon-halogen-aerosol chemistry from the surface to the mesopause (Bey et al., 2001; Park et al., 2004; Eastham et al., 2014; Wang et al., 2019). As of version 12.9.3, this includes 255 interactive species, 597 gas-phase reactions, 101 heterogeneous reactions, and 153 photolysis reactions. All our simulations use identical prescribed emissions from the CMIP6 experiments as described in Sect. 5.1 (as opposed to the default GEOS-Chem anthropogenic inventories) and natural emissions vary with each model
meteorology as described in Sect. 5.2. Volcanic emissions of $SO_2$ from 2005 were used in all simulation years.

### 6.2.1 Hydroxyl Radical

Table 5 assesses hydroxyl radical (OH) concentrations in the model by comparing the simulated lifetime of relatively long-lived molecules whose main sink is tropospheric OH with observational constraints. It is common for atmospheric models to be biased high with respect to OH in these observational constraints (i.e., low with respect to lifetime, e.g., Naik et al.,
2013; Voulgarakis et al., 2013), which is true as well in our three simulations. However, the lifetime of methyl chloroform ($CH_3CCl_3$) and methane in the E2.1 simulations is statistically consistent with the low-end of the observed constraint of $6.0^{+0.5}_{-0.4}$ yr and $10.2^{+0.9}_{-0.7}$ yr from Prinn et al. (2005), respectively. All simulations fall within the multi-model estimates of a tropospheric methane lifetime of $10.2 \pm 1.7$ yr (Fiore et al., 2009) and $9.8 \pm 1.6$ yr (Voulgarakis et al., 2013) but fall short of the observationally derived estimates of $11.2 \pm 1.3$ yr from Prather et al. (2012). In all simulations, the E2.1 simulations
better match the observational constraints. The E2.1-driven simulations in the tropics have thinner overhead ozone columns (see Sect. 6.2.4), greater free-tropospheric water vapor abundances, and greater lightning NO emissions (see Fig. 8), all of which promote increased OH (e.g., Murray et al., 2014).





**Table 6.** Tropospheric total reactive nitrogen $(NO_y)^a$ family budget for 2005-2014 C.E.

| | MERRA-2 | E2.1 (Nudged) | E2.1 |
|---|---|---|---|
| **Burden (Tg N)** | **0.91 ± 0.01** | **0.95 ± 0.02** | **0.93 ± 0.02** |
| NO | 0.03 ± 0.00 (3 %) | 0.02 ± 0.00 (2 %) | 0.02 ± 0.00 (2 %) |
| $NO_2$ | 0.09 ± 0.00 (10 %) | 0.09 ± 0.00 (9 %) | 0.09 ± 0.00 (10 %) |
| $HNO_3$ | 0.26 ± 0.00 (29 %) | 0.22 ± 0.00 (23 %) | 0.21 ± 0.01 (23 %) |
| PAN | 0.25 ± 0.00 (27 %) | 0.27 ± 0.00 (28 %) | 0.24 ± 0.00 (26 %) |
| $RONO_2{}^b$ | 0.16 ± 0.00 (18 %) | 0.19 ± 0.00 (20 %) | 0.19 ± 0.01 (20 %) |
| Other | 0.13 ± 0.01 (14 %) | 0.17 ± 0.01 (18 %) | 0.17 ± 0.01 (18 %) |
| **Sources (Tg N yr$^{-1}$)** | **62 ± 1.0** | **67 ± 1.2** | **67 ± 1.4** |
| Land Fuel Combustion | 35 ± 1.1 (56 %) | 35 ± 1.1 (52 %) | 35 ± 1.1 (52 %) |
| Shipping | 7.3 ± 0.5 (12 %) | 7.3 ± 0.5 (11 %) | 7.3 ± 0.5 (11 %) |
| Aircraft | 0.9 ± 0.0 (1 %) | 0.9 ± 0.0 (1 %) | 0.9 ± 0.0 (1 %) |
| Lightning | 5.8 ± 0.1 (9 %) | 7.5 ± 0.2 (11 %) | 6.2 ± 0.2 (9 %) |
| Open Fires | 4.0 ± 0.4 (6 %) | 4.0 ± 0.4 (6 %) | 4.0 ± 0.4 (6 %) |
| Soil Microbial Activity | 6.1 ± 0.3 (10 %) | 8.0 ± 0.1 (12 %) | 8.4 ± 0.2 (13 %) |
| Transport from Stratosphere | 3.3 ± 0.2 (5 %) | 4.6 ± 0.1 (7 %) | 5.3 ± 0.3 (8 %) |
| **Sinks (Tg N yr$^{-1}$)** | **62 ± 0.8** | **67 ± 0.7** | **67 ± 0.9** |
| Dry Deposition | 34 ± 0.6 (54 %) | 38 ± 0.5 (57 %) | 38 ± 0.6 (57 %) |
| Wet Deposition | 29 ± 0.5 (46 %) | 29 ± 0.5 (43 %) | 29 ± 0.7 (43 %) |
| **Lifetime (d)** | **5.3 ± 0.05** | **5.2 ± 0.04** | **5.0 ± 0.04** |

$^a NO_y \equiv NO + NO_2 + NO_3 + HNO_2 + HNO_3 + HNO_4 + PAN + 2 \cdot N_2O_5 + NIT + NITs + BrNO_2 + BrNO_3 + ClNO_2 + ClNO_3 + INO + IONO_2 + IONO + RONO_2$.

$^b RONO_2 \equiv$ ETHLN + ETNO3 + HONIT + ICN + 2 IDN + IHN1 + IHN2 + IHN3 + IHN4 + INDIOL + INPB + INPD + IONITA + IPRNO3 + ITCN + ITHN + MCRHN + MENO3 + MONITA + MONITS + MONITU + MPAN + MPN + MVKN + NPRNO3 + PPN + PROPNN + PRPN + R4N2.

### 6.2.2 Oxidized Nitrogen

Table 6 gives the tropospheric budget for total reactive nitrogen $(NO_y)$ in all three simulations, which includes $NO_x$ $(\equiv NO$
$+ NO_2)$ and its longer-lived reservoir species such as nitric acid $(HONO_2)$ and peroxyacetylnitrate $(CH_3C(O)OONO_2;$ PAN). The E2.1 simulations have greater total sources of $NO_y$ because of the larger natural sources from lightning and soils (see Sect. 5.2) and a greater flux of $NO_y$ transported from the stratosphere from the products of nitrous oxide $(N_2O)$ oxidation. The $NO_y$ speciation between family members is largely consistent between the simulations. The lifetime of $NO_y$ in the nudged E2.1 simulation is longer than in the free-running E2.1 simulation, reflecting the reduction in that simulation's global mean
precipitation flux.



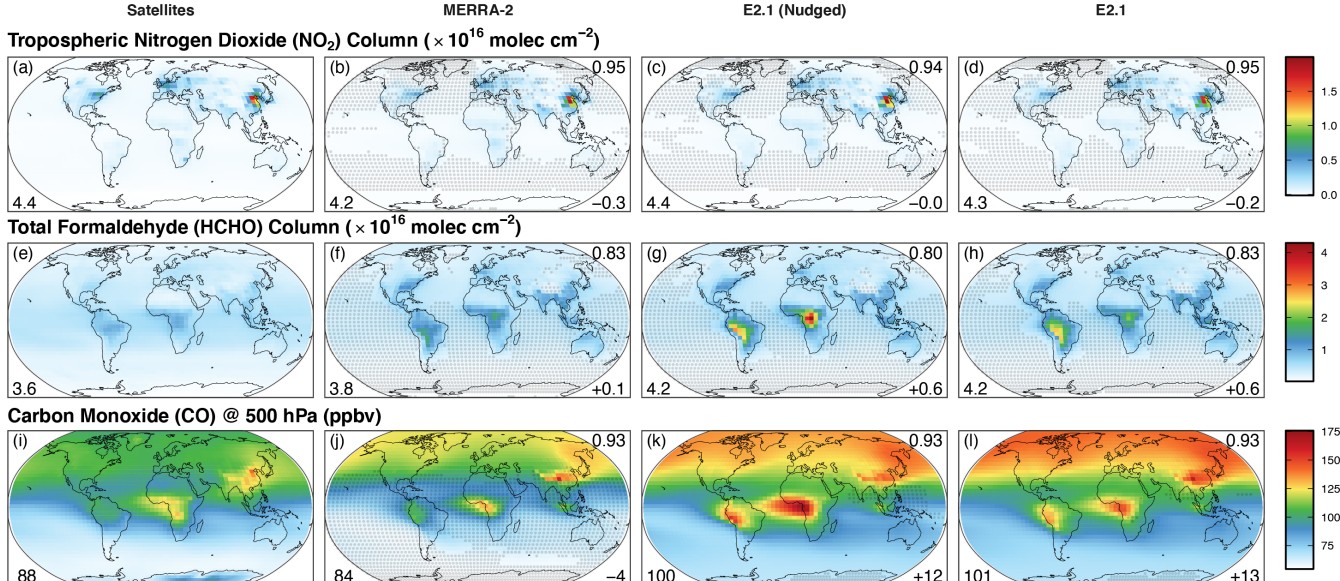

**Figure 15.** Annual average tropospheric columns of nitrogen dioxide (in $10^{16}$ molec cm$^{-2}$; top row), total columns of formaldehyde (in $10^{16}$ molec cm$^{-2}$; middle row), and carbon monoxide mixing ratio at 500 hPa (in ppbv; bottom row) for 2005-2014 C.E.. The left column from top to bottom shows the respective observations from OMI (Krotkov et al., 2017; González Abad et al., 2015) and AIRS (Tian et al., 2020). The right three columns show equivalent values determined from GEOS-Chem driven by MERRA-2, E2.1 nudged to MERRA-2, and free-running E2.1 meteorology. Gray dots show locations where the simulated values are statistically different from the observations with respect to interannual variability ($p$-value $< 0.05$; $n$ = 10 yr). The number in the lower left of each panel shows the global mean value. The number in each model panel's top right shows the pattern correlation ($R$) between the simulated and observed values. The number in the lower left shows the mean bias of the model with respect to the observations. The tropopause was determined in the simulations using the thermal lapse rate for comparison with satellite tropospheric products.

Figure 15a-d compares the spatial distribution of tropospheric columns of nitrogen dioxide (NO$_2$) for 2005-2014 C.E. from version 3 of the OMNO2d product from the Ozone Monitoring Instrument (OMI) on the Aura satellite (Krotkov et al., 2017, doi:10.5067/Aura/OMI/DATA3007) to the three simulations. Fig. S61 in the supplementary materials provides seasonal detail. The simulations have been sampled at the satellite's overpass time and the tropopause was determined online within the model following the thermal definition is used to calculate the partial columns. All simulations well-reproduce the spatial distribution of tropospheric NO$_2$ (all $R \geq 0.94$) and have small global mean low biases, but statistically disagree with the satellite product in the subtropical latitudes. The subtropical disagreement is potentially due to uncertainties in the tropopause height between the various products. Tropospheric columns match over East Asia during this period but are underestimated over North America and Europe.





**Table 7.** Biogenic emissions and tropospheric lifetimes of select non-methane hydrocarbon (NMHC) species for 2005-2014 CE.

|  | MERRA-2 | E2.1 (Nudged) | E2.1 |
|---|---|---|---|
| **Acetone** | | | |
| Terrestrial Source (Tmol yr$^{-1}$) | 0.75 ± 0.0 (47 %) | 0.89 ± 0.0 (49 %) | 0.91 ± 0.0 (50 %) |
| Marine Source (Tmol yr$^{-1}$) | 0.77 ± 0.0 (49 %) | 0.86 ± 0.0 (47 %) | 0.83 ± 0.0 (46 %) |
| Lifetime (d) | 83 ± 2 | 83 ± 1 | 83 ± 2 |
| **Methanol** | | | |
| Terrestrial Source (Tmol yr$^{-1}$) | 2.62 ± 0.07 (86 %) | 3.26 ± 0.05 (88 %) | 3.37 ± 0.15 (88 %) |
| Marine Source (Tmol yr$^{-1}$) | 0.11 ± 0.00 (4 %) | 0.13 ± 0.00 (3 %) | 0.12 ± 0.00 (3 %) |
| Lifetime (d) | 6.6 ± 0.07 | 8.7 ± 0.07 | 8.5 ± 0.06 |
| **Ethanol** | | | |
| Terrestrial Source (Tmol yr$^{-1}$) | 0.49 ± 0.01 (85 %) | 0.60 ± 0.01 (87 %) | 0.61 ± 0.03 (87 %) |
| Lifetime (d) | 4.1 ± 0.05 | 5.4 ± 0.09 | 5.3 ± 0.07 |
| **Acetaldehyde** | | | |
| Terrestrial Source (Tmol yr$^{-1}$) | 0.51 ± 0.01 (21 %) | 0.63 ± 0.01 (24 %) | 0.64 ± 0.04 (25 %) |
| Marine Source (Tmol yr$^{-1}$) | 1.35 ± 0.02 (56 %) | 1.49 ± 0.02 (56 %) | 1.45 ± 0.01 (55 %) |
| Lifetime (d) | 3.4 ± 0.05 | 3.4 ± 0.05 | 3.5 ± 0.05 |
| **Lumped ≥ C$_3$ alkenes** | | | |
| Terrestrial Source (Tmol yr$^{-1}$) | 0.51 ± 0.01 (28 %) | 0.60 ± 0.01 (32 %) | 0.62 ± 0.03 (32 %) |
| Lifetime (d) | 1.8 ± 0.02 | 1.8 ± 0.02 | 2.0 ± 0.02 |
| **Isoprene** | | | |
| Terrestrial Source (Tmol yr$^{-1}$) | 5.2 ± 0.2 (100 %) | 7.4 ± 0.2 (100 %) | 7.3 ± 0.5 (100 %) |
| Lifetime (h) | 13.8 ± 0.5 | 12.2 ± 0.4 | 11.5 ± 0.7 |
| **$\alpha$-pinene, $\beta$-pinene, Sabinene, Carene** | | | |
| Terrestrial Source (Tmol yr$^{-1}$) | 0.57 ± 0.02 (99 %) | 0.72 ± 0.02 (99 %) | 0.74 ± 0.05 (99 %) |
| Lifetime (h) | 3.1 ± 0.09 | 2.3 ± 0.05 | 2.4 ± 0.08 |
| **Other Monoterpenes** | | | |
| Terrestrial Source (Tmol yr$^{-1}$) | 0.33 ± 0.01 (100 %) | 0.42 ± 0.01 (100 %) | 0.43 ± 0.02 (100 %) |
| Lifetime (h) | 2.8 ± 0.08 | 2.0 ± 0.04 | 2.2 ± 0.08 |

Annual mean and standard deviation. The percentage of total emission is given per biogenic source.





**Table 8.** Tropospheric carbon monoxide (CO) budget for 2005-2014 CE.

|  | MERRA-2 | E2.1 (Nudged) | E2.1 |
|---|---|---|---|
| **Burden (Tg)** | **350 ± 10** | **420 ± 10** | **430 ± 10** |
| **Sources (Tg yr$^{-1}$)** | **2440 ± 30** | **2700 ± 40** | **2700 ± 80** |
| Direct Emission | 930 ± 30 (38 %) | 930 ± 30 (34 %) | 930 ± 30 (34 %) |
| Fossil Fuels & Industry | 610 ± 10 (25 %) | 610 ± 10 (22 %) | 610 ± 10 (22 %) |
| Open Fires | 320 ± 30 (13 %) | 320 ± 30 (12 %) | 320 ± 30 (12 %) |
| Chemical Production | 1510 ± 20 (62 %) | 1770 ± 40 (66 %) | 1770 ± 70 (66 %) |
| Methane Oxidation | 800 ± 10 (33 %) | 800 ± 20 (30 %) | 810 ± 20 (30 %) |
| NMHC Oxidation | 700 ± 20 (29 %) | 970 ± 20 (36 %) | 960 ± 50 (36 %) |
| **Sinks (Tg yr$^{-1}$)** | **2440 ± 40** | **2700 ± 40** | **2700 ± 70** |
| Chemical Loss | 2400 ± 30 (99 %) | 2670 ± 30 (99 %) | 2660 ± 70 (99 %) |
| Transport to Stratosphere | 40 ± 10 (1 %) | 30 ± 10 (1 %) | 30 ± 20 (1 %) |
| **Lifetime (d)** | **53.6 ± 0.5** | **57.9 ± 0.8** | **59.5 ± 0.8** |

Annual mean and standard deviation. The percentage of the total is given per source and sink.

### 6.2.3 Reduced Carbon

Table 7 gives the tropospheric emissions and lifetimes for key reduced carbon species in all three simulations. Species primarily emitted from terrestrial plants such as isoprene and monoterpenes have higher emission rates in E2.1 due to the higher diffuse radiation fluxes than in MERRA-2 (see Sect. 4 and Sect. 5.2). Species that are primarily lost via oxidation by OH, such as isoprene, have shorter lifetimes in E2.1 due to the higher OH abundances (see Table 5). In contrast, soluble species such as methanol have longer atmospheric lifetimes in E2.1 due to the lower large-scale stratiform precipitation rates in E2.1 relative to MERRA-2.

Figure 15e-h compares total columns of formaldehyde (HCHO) from version 3 of the OMHCHOd product from OMI on the Aura satellite (González Abad et al., 2015, doi:10.5067/Aura/OMI/DATA3010) to the three simulations for 2005-2014 C.E. Fig. S62 in the supplementary materials provides seasonal detail. The simulations have been sampled at the overpass time of the satellite. Formaldehyde is a common product of hydrocarbon oxidation. Despite higher mean values over the continents, all three simulations are statistically consistent with respect to the large amount of interannual variability in the simulated and observed HCHO columns. There is strong spatial correlation between the simulations and the satellite product (all $R \geq 0.8$). Terrestrial columns simulated by the E2.1 simulations are higher than the MERRA-2 simulations, reflecting the higher biogenic emissions. All simulations underestimate HCHO columns over the remote ocean aside from the continental outflows of North America and Asia and over the Arctic.



Table 8 gives the tropospheric budget for carbon monoxide (CO) in all three simulations. The direct emissions of CO from anthropogenic and biomass burning sources between the three simulations are identical by experimental design. However, the chemical source of CO from methane and non-methane hydrocarbon oxidation is higher in the E2.1 simulations due to the higher OH abundances (Table 5). This is balanced by the increased chemical loss of CO by OH. The influence of OH on CO
production from short-lived non-methane hydrocarbon species seems to outweigh the influence on CO loss, and consequently there are slightly longer CO tropospheric lifetimes in the E2.1 simulations.

Figure 15i-l compares CO mixing ratios at 500 hPa from version 7 of the AIRX3STD product from the Atmospheric Infrared Sounder (AIRS) on the Aqua satellite (Tian et al., 2020, doi:10.5067/8XB4RU470FJV) to the three simulations for 2005-2014 C.E. Fig. S63 in the supplementary materials provides seasonal detail. The simulations have been sampled at the overpass
time of the satellite. All simulations have strong spatial correlation with the observations ($R = 0.93$). CO in the free troposphere is higher in the E2.1 simulations, consistent with its longer lifetime (Table 8). The E2.1 simulations are globally biased high by 15 % compared to the AIRS values. However, they are statistically consistent almost everywhere with respect to the large amount of interannual variability in the observations and simulations. In contrast, the MERRA-2 simulations are globally biased low by 5 % and significantly so throughout the tropics and Southern Hemisphere. All models underestimate the curious
enhancement of CO seen in the AIRS climatology over Antarctica.

### 6.2.4 Ozone

Table 9 gives the tropospheric budget for the odd-oxygen family ($O_x \equiv O_3$ + rapid cycling species) for all three simulations. The individual budget terms are all are consistent with the range of reported values from the Tropospheric Ozone Assessment Report (TOAR) multi-model assessment (see Fig. 3 of Young et al., 2018), the CMIP6 models that performed interactive
tropospheric chemistry (Griffiths et al., 2021), as well as the last extensive tropospheric ozone budget evaluation within the standard GEOS-Chem model (Hu et al., 2017). The E2.1 simulations are on the high end of the previously reported values due to the higher tropopause height in those simulations; the upper troposphere and lower stratosphere regions contribute strongly to each $O_x$ budget term due to the rapidly increasing abundances of ozone with altitude there. We point out that the stratosphere-to-troposphere flux calculated using the "residual method" of the other budget terms yields consistent results
when we track the mass flux of ozone across the dynamic tropopause in the model. Relative to the earlier GCAP and ICECAP simulations (Murray et al., 2014), the transport of ozone from the stratosphere is dramatically improved.

Figure 16 evaluates the zonal and seasonal distribution of ozone versus *in situ* measurements. We use the ozonesonde measurements archived by the World Ozone and Ultraviolet Radiation Data Centre (WOUDC) of the World Meteorological Organization/Global Atmosphere Watch Program (WMO/GAW). The data were accessed on Nov 4, 2019, from doi:10.14287/10000008.
All models fall within the variability of the measurements, except the free troposphere of the northern extratropics, where the simulations are biased low, especially during the summer months. The MERRA-2 simulations better reproduce the seasonality of ozone in the tropical free and upper troposphere, likely reflecting the climatologically constrained lightning NO source in that version (see Sect. 5.2).





**Table 9.** Tropospheric odd-oxygen ($O_x$)[a] family budget for 2005-2014 C.E.

|  | MERRA-2 | E2.1 (Nudged) | E2.1 |
|---|---|---|---|
| **Burden (Tg $O_x$)** | **317 $\pm$ 4.3** | **338 $\pm$ 4.4** | **368 $\pm$ 4.4** |
| O$_3$ | 315 $\pm$ 4.2 | 336 $\pm$ 4.4 | 366 $\pm$ 4.4 |
| Other | 2.4 $\pm$ 0.03 | 2.4 $\pm$ 0.03 | 2.3 $\pm$ 0.04 |
| **Sources (Tg $O_x$ yr$^{-1}$)** | **5200 $\pm$ 40** | **5620 $\pm$ 70** | **5700 $\pm$ 130** |
| Transport from Stratosphere | 580 $\pm$ 20 | 620 $\pm$ 30 | 870 $\pm$ 40 |
| Chemical Production | 4710 $\pm$ 50 | 5080 $\pm$ 80 | 4920 $\pm$ 120 |
| **Sinks (Tg $O_x$ yr$^{-1}$)** | **5200 $\pm$ 42** | **5620 $\pm$ 70** | **5700 $\pm$ 130** |
| Chemical Loss | 4240 $\pm$ 41 | 4550 $\pm$ 70 | 4660 $\pm$ 130 |
| Dry Deposition | 891 $\pm$ 9.6 | 1000 $\pm$ 9.6 | 975 $\pm$ 7.6 |
| O$_3$ | 795 $\pm$ 8.2 | 891 $\pm$ 9.2 | 867 $\pm$ 8.5 |
| Other | 96 $\pm$ 1.5 | 112 $\pm$ 1.1 | 108 $\pm$ 2.6 |
| Wet Deposition | 70 $\pm$ 0.9 | 67 $\pm$ 0.9 | 65 $\pm$ 1.9 |
| Stratiform | 54 $\pm$ 0.9 | 10 $\pm$ 0.3 | 10 $\pm$ 0.5 |
| Convective | 16 $\pm$ 0.3 | 57 $\pm$ 0.9 | 55 $\pm$ 1.8 |
| **Lifetime (d)** | **22.3 $\pm$ 0.26** | **22.0 $\pm$ 0.40** | **23.6 $\pm$ 0.54** |

[a] $O_x \equiv O_3 + O(^3P) + O(^1D) + NO_2 + 2 \cdot NO_3 + PAN + PPN + MPAN + HNO_4 + 3 \cdot N_2O_5 + HNO_3 + BrO + HOBr + BrNO_2 + 2 \cdot BrNO_3 + MPN + ETHLN + MVKN + MCRHN + MCRHNB + PROPNN + R4N2 + PRN1 + PRPN + R4N1 + HONIT + MONITS + MONITU + OLND + OLNN + IHN1 + IHN2 + IHN3 + IHN4 + INPB + INPD + ICN + 2 \cdot IDN + ITCN + ITHN + ISOPNOO1 + ISOPNOO2 + INO2B + INO2D + INA + IDHNBOO + IDHNDOO1 + IDHNDOO2 + IHPNBOO + IHPNDOO + ICNOO + 2IDNOO + MACRNO_2 + ClO + HOCl + ClNO_2 + 2 \cdot ClNO3 + 2 \cdot Cl_2O_2 + 2 \cdot OClO + IO + HOI + IONO + 2 \cdot IONO_2 + 2 \cdot OIO + 2 \cdot I_2O_2 + 3 \cdot I_2O_3 + 4 \cdot I_2O_4$.
A molar mass of 48 g is assumed for $O_x$.

Figure 17 evaluates spatial distributions of ozone in the three simulations against surface *in situ* and satellite observations.
The top row shows the annual average surface ozone mixing ratio in ppbv ($\equiv$ nmol mol$^{-1}$) from our simulations versus the gridded mean non-urban surface ozone product from the Tropospheric Ozone Assessment Report (TOAR) (Schultz et al., 2017, 10.1594/PANGAEA.876108). The middle row shows tropospheric columns of ozone (TCO; in Dobson Units) versus the joint Ozone Monitoring Instrument (OMI) and Microwave Limb Sounder (MLS) product from the Aura satellite (Ziemke et al., 2006). The bottom row shows total ozone columns (TOC; in Dobson Units) versus the OMDOAO3e product from
OMI (Dobber et al., 2006, doi:10.5067/Aura/OMI/DATA3005). All observational products have been aggregated to model resolution for comparison. In the case of the satellite products, we sampled the model at the overpasses' time and location. We use the online thermal lapse rate tropopause to calculate TCO. Seasonal details are given in Figs. S64-S66 of the supplementary materials.

All simulations are biased high by 15-17 % with respect to surface ozone, mostly driven by the eastern North American
data. The E2.1 simulations are especially higher over the Amazon than either MERRA-2 or the observations. Comparisons of



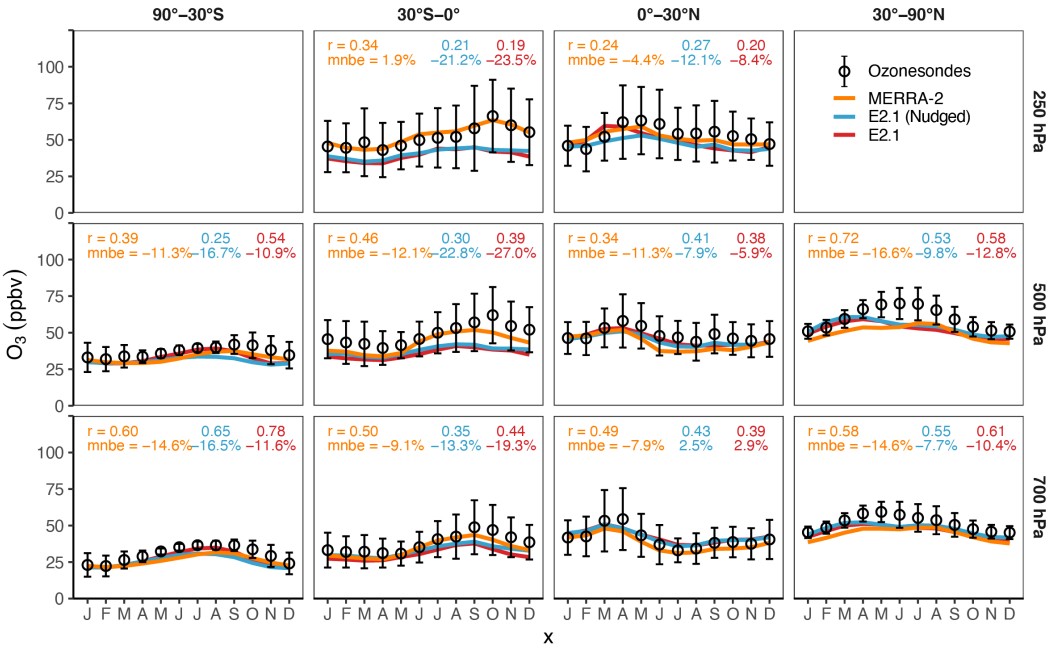

**Figure 16.** Comparison of the annual cycle of ozone for 2005-2014 C.E. between ozonesonde observations (black circles) and the MERRA-2 (solid orange line), E2.1 nudged to MERRA-2 (blue line), and E2.1 (red line) simulations. Model and observational data were grouped into four latitude bands (90°S to 30°S, 30°S to 0°, 0°S to 30°N and 30°S to 90°N) and sampled at three altitudes (700, 500 and 250 hPa), with the models sampled at locations and months of the ozonesonde measurements before averaging together. Error bars on the observations indicate the average interannual standard deviation for each group of observations. The correlation ($r$) and mean normalized bias error (mnbe) for the MERRA-2 (orange), E2.1 nudged to MERRA-2 (blue), and E2.1 (red) means versus the observations are also indicated in each panel.

TCO to OMI/MLS are sensitive to uncertainties in the tropopause location in the satellite product versus the models (Griffiths et al., 2021). The E2.1 TCO are globally higher than the OMI/MLS product by 12 %, reflecting the lower tropopause pressures. However, the only locations in which it is statistically different regarding interannual variability are over the tropical oceans, where it is biased low. This likely reflects the more vigorous convection in E2.1 that leads to ozone destruction (e.g., Murray

et al., 2013). Tropospheric columns in MERRA-2 match the global mean from OMI/MLS but also underestimate the western Pacific and additionally underestimate northern extratropical ozone. All simulations underestimate tropospheric columns in the southern extratropics with respect to OMI/MLS. The models all show excellent agreement with respect to total ozone columns with small positive mean global biases of 4 % and high pattern correlation (all $R \geq 0.96$). Total ozone in the tropics is higher in MERRA-2 than E2.1, consistent with differences in the rate of vertical ascent in the tropical pipe implied by the stratospheric

age of air comparison (see Sect. 6.1.3). The E2.1 simulation overestimates Antarctic ozone relative to MERRA-2 or the nudged simulation, although not significantly compared to interannual variability.

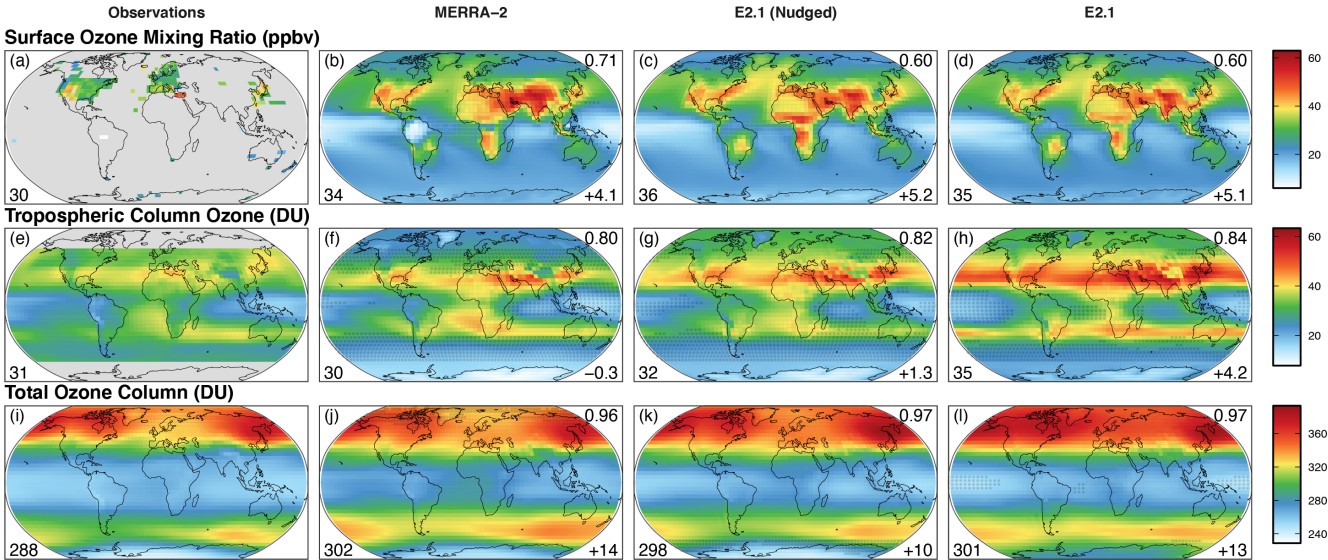

**Figure 17.** Annual average surface ozone mixing ratio in ppbv ($\equiv$ nmol mol$^{-1}$; top row), tropospheric columns of ozone (TCO; in Dobson Units; middle row), and total ozone columns (TOC; in Dobson Units; bottom row) for 2005-2014 C.E. The left column from top to bottom shows observations from TOAR (Schultz et al., 2017), OMI/MLS (Ziemke et al., 2006) and OMI (Dobber et al., 2006), respectively. The right three columns show equivalent values determined from GEOS-Chem driven by MERRA-2, E2.1 nudged to MERRA-2, and free-running E2.1 meteorology. Gray dots show locations where the simulated values are statistically different from the observations with respect to interannual variability ($p$-value $< 0.05$; $n = 10$ yr). The number in the lower left of each panel shows the global mean value. The number in each model panel's top right shows the pattern correlation ($R$) between the simulated and observed values. The number in the lower left shows the mean bias of the model with respect to the observations. The tropopause was determined in the simulations using the thermal lapse rate for comparison with satellite tropospheric products.

### 6.2.5 Particulate Matter

Figure 18 evaluates the spatial distribution of particulate matter in the three simulations against satellite observations. The top row shows the simulated concentration of fine particulate matter under 2.5 microns (PM$_{2.5}$) in $\mu$g m$^{-3}$ versus the historical re-
construction from Hammer et al. (2020). The middle row shows the total AOT at 550 nm (unitless) in the simulations versus the Combined Dark Target and Deep Blue Aerosol Optical Thickness (AOT) at 0.55 micron from collection 6.1 of the MODerate resolution Imaging Spectroradiometer (MODIS) MYD08 product from the Aura satellite (doi:10.5067/MODIS/MYD08_M3.061). The bottom row shows the total column of sulfur dioxide (SO$_2$) in Dobson Units in the simulations versus the second public re-lease of version 3 of the OMSO2e product from OMI on the Aura satellite (Li et al., 2020, doi:10.5067/Aura/OMI/DATA3008).
The model has been sampled at the time of the Aura overpasses for comparison to the satellite products. Figures S67-S68 of the supplementary materials provide seasonal detail for AOT and the SO$_2$ columns.



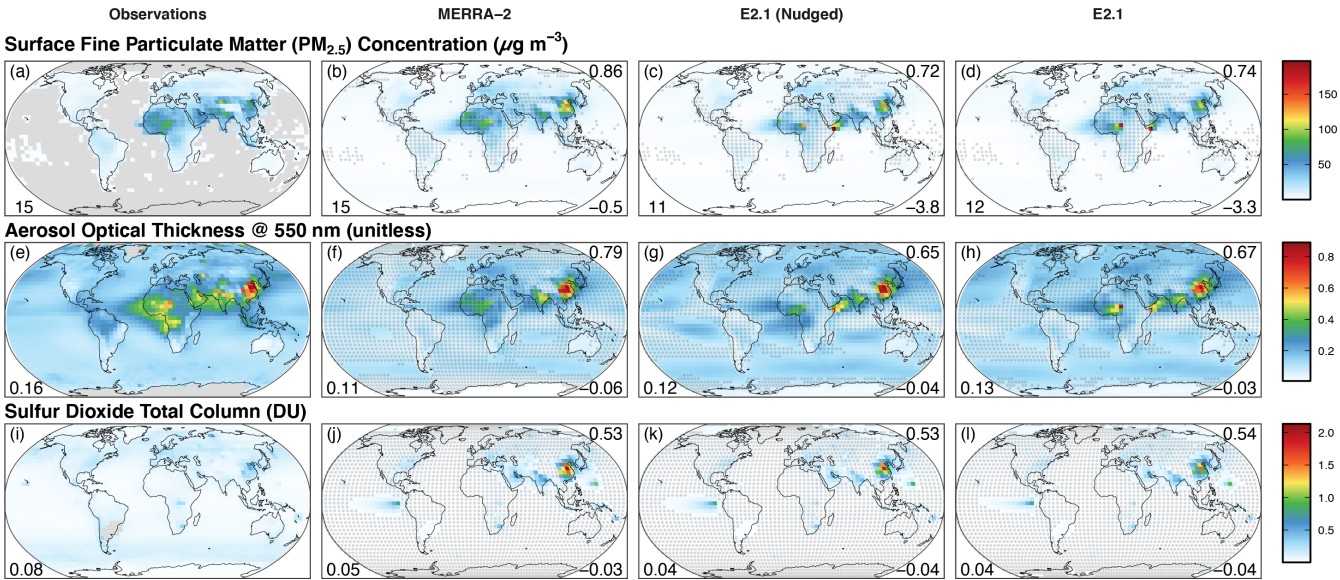

**Figure 18.** Annual average surface concentration of fine particulate matter (PM$_{2.5}$; in $\mu$g m$^{-3}$; top row), aerosol optical thickness at 550 nm (unitless; middle row), and total column of sulfur dioxide (in Dobson Units; bottom row). The left column from top to bottom shows observations from Hammer et al. (2020), Aura MODIS, and OMI (Li et al., 2020), respectively. The right three columns show equivalent values determined from GEOS-Chem driven by MERRA-2, E2.1 nudged to MERRA-2, and free-running E2.1 meteorology. Gray dots show locations where the simulated values are statistically different from the observations with respect to interannual variability ($p$-value $< 0.05$; $n = 10$ yr). The number in the lower left of each panel shows the global mean value. The number in each model panel's top right shows the pattern correlation ($R$) between the simulated and observed values. The number in the lower left shows the mean bias of the model with respect to the observations.

Surface PM$_{2.5}$ agrees between the simulations and Hammer et al. (2020) over heavily industrialized regions. However, we note that the Hammer et al. (2020) product used ratios of surface PM$_{2.5}$ to AOT from GEOS-Chem to generate their proxy reconstruction from satellite AOT measurements, so it is not an entirely independent comparison. Surface concentrations are underestimated almost everywhere else in all simulations, especially in regions heavily influenced by mineral dust and biomass burning. The same story is reflected in the MODIS AOT, except the stronger sea-salt aerosol emission in the E2.1 simulations yielding better comparison in the Southern Ocean and portions of the Pacific. Simulated total columns of sulfur dioxide are underestimated everywhere in all simulations except over East Asia and the locations of the major volcanic eruptions of 2005 (Sierra Negra in the Galápagos and Anatahan of the Northern Mariana Islands).

## 7 Summary

This manuscript described and evaluated Version 2.0 of the Global Change and Air Pollution (GCAP) chemical-transport model framework.





GCAP 2.0 is a one-way offline coupling between the E2.1 version of the NASA GISS GCM frozen for the CMIP6 experiments (Kelley et al., 2020; Miller et al., 2021) and the GEOS-Chem 3-D chemical-transport model (http://www.geos-chem.org; Bey et al., 2001). Additional sub-daily diagnostics were added to E2.1 to archive the same fields as the MERRA-2 reanalysis product (Gelaro et al., 2017) that is normally used to drive GEOS-Chem. We then re-performed one of the atmosphere-only members of the E2.1 contributions to the CMIP6 ensemble, archiving the meteorology necessary for driving GEOS-Chem. The E2.1 meteorology is available at 2° latitude by 2.5° longitude and for 40 vertical layers ranging from the surface to 0.1 hPa. At publication time, meteorology is available for the preindustrial (1851-1860 C.E.) and recent past (2001-2014 C.E.), including a recent-past simulation nudged to MERRA-2 to assist users in comparing with observations. Also available is meteorology for the near future (2040-2049 C.E.) and the end-of-the-century (2090-2099 C.E.) for seven future "Shared Socioeconomic Pathway" (SSP) scenarios ranging from extreme mitigation (SSP1-1.9) to extreme warming (SSP5-8.5). In addition, the CMIP6 emissions and surface boundary conditions (Hoesly et al., 2018; van Marle et al., 2017; Riahi et al., 2017; Meinshausen et al., 2017; Gidden et al., 2019) have been processed for input into GEOS-Chem. GCAP 2.0 is operational in all current variants of the GEOS-Chem model, with all GCClassic run directories and input files provided. All GCAP 2.0 input data is publicly served at http://atmos.earth.rochester.edu/input/gc/ExtData/.

The meteorology was evaluated by comparing to both the original simulation and the MERRA-2 reanalysis for the recent past. Surface air in the repeat simulation is slightly warmer than the original run due to increased calls to the radiation code necessary for archiving shortwave fluxes for input to GEOS-Chem. The E2.1 climatology in the recent past largely agrees with the MERRA-2 climatology for that period with the primary difference being in the relative amount of precipitation in stratiform versus convective clouds as well as a higher tropopause height in E2.1. Emissions that respond to meteorology in GEOS-Chem are slightly higher in the E2.1-driven simulations, including biogenic emissions from terrestrial plants, the lightning and soil microbial sources of reactive nitrogen, and sea-salt evasion. The dust mobilization parameterization was found to be extremely sensitive to resolution and meteorology, and scaling factors have been determined to constrain the global source.

Model physics and transport were evaluated using simulations and observations of sulfur hexaflouride and radionuclides. In all cases, transport is substantially improved over the original GCAP, and the E2.1-driven simulations perform comparably to the MERRA-2-driven simulations. Most importantly, whereas age of air remains too young in both E2.1 and MERRA-2, the stratosphere-to-troposphere mass flux now yields comparable values, with consistent stratosphere-to-troposphere fluxes of ozone with multi-model means. This is a major improvement over the previous versions of GCAP (Wu et al., 2007; Murray et al., 2014). However, we urge users to be cautious when using these fields for stratospheric chemistry-climate applications. Future simulations will provide CMIP6 meteorology from the 102-layer version of the GISS GCM (E2.2; Rind et al., 2020), which will be better suited for studies of the middle atmosphere; E2.2 includes an interactive quasi-biennial oscillation and improved polar vortex variability including sudden warmings, which could contribute additional dynamical variability that GEOS-Chem may otherwise not see (Orbe et al., 2020b).

Lastly, the chemistry of the model using the CMIP6 emissions and the different meteorologies was evaluated against a suite of satellite products and *in situ* observations. The E2.1-driven simulations have higher OH and therefore more accurate methyl chloroform and methane lifetimes. Greater biogenic fluxes and higher OH yield greater abundances of oxidation products





(e.g., CO) in the E2.1-driven simulations, improving comparison with observations in the southern hemisphere. However, the MERRA-2-driven simulations have a superior representation of free-tropospheric ozone, likely due to the constrained lightning
seasonality and distribution as well as the more realistic tropopause pressure in those simulations. All simulations underestimate particulate matter abundances outside of industrialized areas and AOT in most places. Otherwise, model performance was very similar in all simulations.

*Code and data availability.* GCAP 2.0 in version 13.0.0 of GEOS-Chem is available at doi:10.5281/zenodo.4783680, doi:10.5281/zenodo.4783686 and doi:10.5281/zenodo.4783703. Source code for generating E2.1 output to drive GEOS-Chem is available at doi:10.5281/zenodo.4783672.
Archived meteorology, emissions and boundary conditions for GCAP 2.0 are hosted online at http://atmos.earth.rochester.edu/input/gc/ExtData/. LTM can generate additional CMIP6 scenarios and time periods upon request.

*Author contributions.* LTM conceived the project, performed the model development, simulation, and evaluation of the E2.1 and GEOS-Chem interface, and wrote the manuscript. EML and LJM provided code edits from ModelE that were updated to E2.1 by LTM. CO provided feedback on the implementation of the new sub-daily diagnostics. MS was a developer of the FlexGrid code in GEOS-Chem. All authors
contributed feedback to the manuscript.

*Competing interests.* The authors declare no competing interests.

*Acknowledgements.* LTM acknowledges funding from NSF grants AGS-1702106 and AGS-2002414. LJM acknowledges funding from NASA grant NNX13AO08G. We thank the entire GISS and GEOS-Chem developer and user communities.





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
