# Peer review of "GCAP 2.0: A global 3-D chemical-transport model framework for past, present, and future climate scenarios"

_Geoscientific Model Development, 2021_

## Author Comment (AC1)

**Reviewer comments shown in black**
Authors' responses shown in blue

**Reviewer #1**

This manuscript presents a description and evaluation of the second generation one-way offline coupling between NASA GISS E2.1 GCM and GEOS-Chem CTM (GCAP 2.0). This is an update of the on the initial GCAP framework, with improved GISS meteorology and GEOS-Chem emissions/chemistry.

The authors include a thorough presentation of the  meteorological fields and emissions, with a comparison of three types of meteorology: MERRA-2, E2.1 nudged to MERRA-2, and E2.1. The model simulations are intercompared and evaluated against observations for atmospheric constituents (O3, NO2, HCHO, CO, PM2.5; lifetimes of CH4 and CH3CCl3; AOD) and transport tracers (SF6 and radionucleides). This enables a detailed evaluation of transport as well as chemistry and deposition. This paper is a substantial contribution to modeling science and provides a new tool for the community. The paper is well writen and well organized. I recommend publication.

We thank the reviewer for their very kind review.

**Technical corrections:**

Please check the figure captions of Figures 15 onwards  as some have a typo "The number in the lower left shows the mean bias of the model with respect to the observations", which should be lower right.

Thank you for catching these typos; they have been corrected in Figs. 15, 17 and 18.

Line 612: The MODIS instrument is onboard the Aqua satellite (not Aura). Also line 616 should be correscted by adding Aqua.

Thank you for catching this typo; it has been corrected throughout the manuscript.

**Reviewer #2**

This manuscript describes the GCAP 2.0 model framework, in which the GEOS-Chem 3D CTM is driven by version E2.1 of the NASA GISS model to simulate past, present, and future atmospheric compositions. The work is built on previous versions of offline coupling between the GISS model and the GEOS-Chem model. This manuscript first explains how the meteorological fields were generated from the GISS E2.1 model, the boundary conditions and emissions taken from latest CMIP6-relevant datasets, and how these data drive the GEOS-Chem model for different time periods. The manuscript then extensively evalute the E2.1 meteorological dataset against a reanalysis product (MERRA-2), the E2.1 product submitted to CMIP6, and an intermediate product in which E2.1 winds were nudged to MERRA-2 winds. The resulting differences in GCAP 2.0-simulated atmospheric composition and chemical diagnostics are also evaluated against observations.

In my view, this paper is extremely well written. It is particularly commendable in its extensive evaluation of the meteorological data sets and chemical diagnostics. I only have a few minor comments. In all, I think this is a great guide for researchers interested in using GCAP 2.0, as well as a great reference for those who are interested in understanding the results from global CTMs. I recommend publication of this manuscript after minor revisions.

We thank the reviewer for their very kind review.

Line 18: "... for the recent past...": This reads awkward after the last sentence of the previous paragraph, which is (largely) not an issue of the recent past.

We have removed the "for the recent past" clause to avoid confusion.

Lines 131-132: This sentence is confusing for readers unfamiliar with GEOS or MERRA-2. Is MERRA-2 a product of GEOS-DAS?

We have added phrasing to emphasize that MERRA-2 is a GMAO/GEOS-DAS product.

"GEOS-Chem (http://www.geos-chem.org) is a global or regional 3-D chemical transport model traditionally driven by assimilated meteorology products produced by the NASA Global Modeling and Assimilation Office (GMAO) Goddard Earth Observing System Data Assimilation System (GEOS-DAS). **These include** the MERRA-2 science product**, which** is generated at…"

Section 5.1.1, lines 259-264: I am confused by this paragraph. So, does GCAP 2.0 use only CEDS, or does it also use a hybrid emission inventory with CEDS superseded by regional inventories?

In GEOS-Chem (and therefore GCAP 2.0), users may easily use any emissions through the HEMCO emissions pre-processor as specified in a single configuration file (HEMCO_Config.rc). The default inventories for GEOS-Chem change frequently with version updates and often include regional overwrites (e.g., the EPA National Emissions Inventory for the United States) and/or scaling factors. Here, we have prepared CMIP6-compliant emissions for users who wish to use GEOS-Chem to perform CMIP6 experiments.

We have added the following sentences to the end of the paragraph to help clarify.

**"When users generate a GCAP 2.0 run directory, the respective CMIP6 emissions and boundary conditions for a historical or future scenario are enabled by default. However, users may always modify the HEMCO configuration to use any emissions they desire**

**(e.g., to use the alternative inventories, regional overwrites and/or scaling factors from the default GEOS-Chem configuration)."**

Line 438: "There is a slight overestimate within the boundary layer and underestimate above ...": overestimate and underestimate of what? Are the authors referring to 222Rn concentration or its vertical gradient? Please revise to imrpove clarity.

Rephrased to clarify.

"There is a slight overestimate **of $^{222}$Rn abundance** within the boundary layer and underestimate above in all of our simulations…"

Figure 18: What is the time period examined in this figure?

We have added the time period to the caption (2005-2014 C.E.).